

# 1 Why models perform differently on particulate matter over East
# 2 Asia? – A multi-model intercomparison study for MICS-Asia III

Jiani Tan[1], Joshua S. Fu[1], Gregory R. Carmichael[2], Syuichi Itahashi[3], Zhining Tao[4], Kan
Huang[1,5], Xinyi Dong[1], Kazuyo Yamaji[6], Tatsuya Nagashima[7], Xuemei Wang[8], Yiming Liu[8],
Hyo-Jung Lee[9], Chuan-Yao Lin[10], Baozhu Ge[11], Mizuo Kajino[12], Jia Zhu[11], Meigen Zhang[11],
Liao Hong[13] and Zifa Wang[11]
[1] Department of Civil and Environmental Engineering, University of Tennessee, Knoxville, TN, 37996, USA
[2] Center for Global and Regional Environmental Research, University of Iowa, Iowa City, IA, 52242, USA
[3] Central Research Institute of Electric Power Industry, Abiko, Chiba, 270-1194, Japan
[4] Universities Space Research Association, Columbia, MD, 21046, USA
[5] Department of Environmental Science and Engineering, Fudan University, Shanghai, 200433, China
[6] Graduate School of Maritime Sciences, Kobe University, Kobe, Hyogo, 658-0022, Japan
[7] National Institute for Environmental Studies, Tsukuba, Ibaraki, 305-8506, Japan
[8] Institute for Environment and Climate Research, Jinan University, Guangzhou, 511443, China
[9] Department of Atmospheric Sciences, Pusan National University, Busan, 609-735, South Korea
[10] Research Center for Environmental Changes Academia Sinica, 11529, Taiwan
[11] Institute of Atmospheric Physics, Chinese Academy of Science, 100029, China
[12] Meteorological Research Institute, Japan Meteorological Agency, 305-0052, Japan
[13] School of Environmental, Science and Engineering, Nanjing University of Information Science & Technology,
Nanjing, 210044, China
*Correspondence to: Joshua S. Fu (jsfu@utk.edu)*
**Abstract.** This study compares the performances of twelve regional chemical transport models
(CTM) from the third phase of Model Inter-Comparison Study for Asia (MICS-Asia III) on
simulating the particulate matter (PM) over East Asia (EA) in 2010. The participating models
include WRF-CMAQ (v4.7.1 and v5.0.2), WRF-Chem (v3.6.1 and v3.7.1), GEOS-Chem, NHM-
Chem, NAQPMS and NU-WRF. Evaluations with ground measurements and satellite data show
that the mean biases of multi-model mean (MMM) are -25 µg m$^{-3}$ (-30%), -7 µg m$^{-3}$ (-15%). -0.7
µg m$^{-3}$ (-19%), -0.05 µg m$^{-3}$ (-3%) and 0.1 µg m$^{-3}$ (12%) for surface $PM_{10}$, $PM_{2.5}$, $SO_4^{2-}$, $NO_3^-$ and
$NH_4^+$ concentrations, respectively. This study investigates four model processes as the possible
reasons for different model performances on PM: (1) Using different natural emissions (i.e. dust
and sea-salt emissions) brings upmost 0.25 µg m$^{-3}$ (70%) of inter-model differences to domain-
average black carbon concentrations at surface layer and 756 ppb (22%) of inter-model differences
to domain-average CO column. Adopting different initial/boundary conditions results in 10-20%
differences in PM concentrations in the center of the simulation domain. (2) Models perform very
differently in the gas-particle conversion of sulphur (S) and oxidized nitrogen (N). The model





differences in sulphur oxidation ratio (50%) is of the same magnitude as that in $SO_4^{2-}$
concentrations. The gas-particle conversion is one the main reasons for different model
performances on fine mode PM. (3) Models without dust emissions/modules can perform well on
$PM_{10}$ at non-dust-affected sites, but largely underestimate (upmost 50%) the $PM_{10}$ concentrations
at dust sites. The implementation of dust emissions/modules in models has largely improved the
model accuracies at dust sites (reduce model bias to -20%). However, both the magnitudes and
distributions of dust pollutions are not fully captured. (4) The amounts of modelled depositions
vary among models by 75%, 39%, 21% and 38% for S wet, S dry, N wet and N dry depositions,
respectively. Large inter-model differences are found in the washout ratios of wet deposition (at
most 170% in India) and dry deposition velocities (general 0.3-2 cm s$^{-1}$ differences over inland
regions). This study investigates the reasons for different model performances on PM over EA and
offers suggestions for future model development.
**1 Introduction**
Atmospheric pollution due to particulate matter (PM) has raised world-wide attention for its
relationship with environmental and public health issues (Fuzzi et al., 2015;Nel, 2005). Fine
particles ($PM_{2.5}$) are associated with cardiovascular and respiratory related cancer and premature
deaths (Hoek and Raaschou-Nielsen, 2014;Knol et al., 2009). Outdoor $PM_{2.5}$ pollution is estimated
to cause 2.1-5.2 million premature deaths worldwide annually (Lelieveld et al., 2015;Rao et al.,
2012;Silva et al., 2013). It accounts for eight percent of global mortality in 2015 and ranks fifth in
the global mortality risk (Cohen et al., 2017). East Asia (EA) has been suffering from severe PM
pollutions due to anthropogenic emissions and natural dust emissions (Akimoto, 2003). China and
India are the top two countries suffering from outdoor air pollutions, which altogether account for
20% of global mortalities caused by $PM_{2.5}$ exposure in 2010 (Lelieveld et al., 2015). The mixing
of dust with anthropogenic pollutants can even enlarge the effects of pollution (Li et al., 2012).
However, the impact evaluation on PM pollution is of high uncertainty due to unclearness in the
toxicity of PM components (Lippmann, 2014) and difficulty in the measurement and prediction of
PM concentrations.
For a better understanding of PM pollution, modelling approach has been adopted to study
the spatial distributions of PM with the aid of measurements. Multi-model ensemble approach,
which interprets modelling results with combined information from several models, has been



proven to increase the reliability of model accuracy (Tebaldi and Knutti, 2007). This method has
been widely used for studies in Europe (Bessagnet et al., 2016;Vivanco et al., 2017) and at global
scales (Lamarque et al., 2013;Galmarini et al., 2017) on air quality issues. The Model Inter-
Comparison Study Asia Phase (MICS-Asia) aims at understanding the air quality issues over EA.
The first phase of MICS-Asia (MICS-Asia I) was carried out in the 1990s with eight regional
chemical transport models (CTMs). The study focused on air pollution issues related to sulphur
(S) (including $SO_2$, $SO_4^{2-}$ and wet $SO_4^{2-}$ deposition). The second phase of MICS-Asia (MICS-Asia
II) was launched in early 2000s with nine CTMs (Carmichael and Ueda, 2008). The study covered
the chemistry and transport of S, nitrogen (N), PM and acid deposition. Multi-model results on
$SO_4^{2-}$, $NO_3^-$ and $NH_4^+$ (SNA) were evaluated with measurements from fourteen sites of Acid
Deposition Monitoring Network in East Asia (EANET) and the Fukue site in Japan. However, a
non-exhaustive evaluation on $PM_{10}$ concentrations in China with scarce datasets left an unclear
view of models' ability in this area, a region recognized as one of the most heavily polluted in EA.
Meanwhile, model results were found with high inconsistencies on simulating both gas and aerosol
phases of S and N (Hayami et al., 2008). Further efforts are needed to investigate the reasons for
model differences to improve model accuracies.
This study compares the performances of twelve regional models participated in the third
phase of MICS-Asia (MICS-Asia III) on simulating PM over EA. Measurements from 54 EANET
site, 86 sites of the Air Pollution Indices (API) and 35 local sites are used for model evaluation to
provide a comprehensive view on model performances. The comparison among models aims at
quantifying the model biases with observation and identifying the reasons for different model
performances. The models involved in this study include WRF-CMAQ (version 4.7.1 and v5.0.2),
WRF-Chem (v3.6.1 and v3.7.1), GEOS-Chem, NHM-Chem, NAQPMS and NU-WRF. The multi-
model mean (MMM) performance on simulating the spatial distributions and monthly variations
of $PM_{10}$, $PM_{2.5}$, SNA and aerosol optical depth (AOD) are evaluated with site and satellite
observations. The evaluation results are demonstrated briefly in sect. 3.1 and details can be found
in supplementary sect. S2. Sections 3.2-3.5 examine the influences of four model processes on
model performances: (1) Source of particles: uncertainties brought by inconsistent model inputs
and initial/boundary conditions (IC/BC) for simulations. (2) Formation of fine particles: model
differences in the gas-particle conversion. (3) Formation of coarse particles: model improvements
by implementing dust emissions/modules on simulating PM and the remaining problems. (4)





Removal processes of particles from the atmosphere: uncertainties lay on the efficiencies of wet
and dry depositions. Section 4 concludes the findings of this study and provides suggestion for
further study.

**2 Methodology**


**2.1 Framework of MICS-Asia**


MICS-Asia is a model intercomparison study with contributions from international modelling
groups to simulate the air quality and deposition over EA. During MICS-Asia I, eight models
simulated the air qualities for January and May of 1993. The study focused on air quality issues
related to S. The multi-model performances on simulating $SO_2$ and $SO_4^{2-}$ concentrations and $SO_4^{2-}$
wet deposition were evaluated with observation from eighteen stations (Carmichael et al., 2002).
A source-receptor relationship of S deposition was developed based on the sensitivity simulations
for seven prescribed receptor regions: Komae, Oki, Fukue, Yangyang, Beijing, Nanjing and
Taichung (Carmichael et al., 2002).
MICS-Asia II was initiated in 2003. Nine regional models simulated the air qualities for
four months (March, July and December of 2001 and March of 2002) to study the chemistry and
transport of air pollutants and acid deposition (Carmichael and Ueda, 2008). All modelling groups
were enforced to use the same emission, the Transport and chemical Evolution over the Pacific
(TRACE-P) emission of 2000, and common BC to facilitate a comparison on the physical and
chemical mechanisms of models. The modelling species expanded to S, N, $O_3$, PM and acid
deposition. Model evaluations and major findings can be found in literature (Carmichael et al.,
2008;Fu et al., 2008;Han et al., 2008;Hayami et al., 2008).
MICS-Asia III is launched in 2010. The simulation time covers the whole year of 2010.
All modelling groups are required to use the prescribed anthropogenic emission inputs (Li et al.,
2017), but the natural emissions such as dust and sea-salt emissions are not defined. Three purposes
are set for this project– topic I: evaluating the strengths and weaknesses of current multi-scale air
quality models in simulating air qualities over EA and providing suggestion to reduce uncertainty
for future simulations, topic II: developing a reliable anthropogenic emission inventory for EA,
topic III: investigating the interaction of aerosol-weather-climate by using online coupled air
quality models. This study focuses on topic I.





## 2.2 Model configurations


There are altogether fourteen modeling groups (M1-M14) participated, but M3 and M9 are not
included in this study due to uncompleted model submission. Table 1 shows the set-ups of the
twelve models. All models have submitted the monthly average concentrations of $PM_{10}$, $PM_{2.5}$ and
SNA at surface layer except $PM_{10}$ from M13 and $NO_3^-$ and $NH_4^+$ from M10. Since the spin-up time
is not required, several models use downscale results from global models as IC and BC. The
impacts of using different emission inputs and IC/BC on model performances are discussed in sect.
3.2. This study examines three model processes related to the formation and removal pathways of
PM:
(1)    Gas and aerosol modules and gas-aerosol equilibrium. One of the main sources of fine-
mode particles (PMF) is newly formed particles from nucleation of vapours. The gas modules in
the models control the formation rate of gases and the aerosol module determines the conversion
between gas and particle phases. This study includes four gas modules: Statewide Air Pollution
Research Center (SAPRC99) (Carter, 2000), Regional Atmospheric Deposition Model (RADM)
(Stockwell et al., 1990), Regional Atmospheric Chemistry Mechanism (RACM) (Stockwell et al.,
1997) and Carbon-Bond Mechanism version Z (CBMZ) (Zaveri and Peters, 1999). Different
modules generally use similar reaction rates in the homogenous production of $SO_4^{2-}$ and $NO_3^-$
aerosols, but have significant differences in the rates of the heterogeneous reactions among $NO_2$,
HONO, $HNO_3$ and $N_2O_5$ (supplementary fig. S1). The aerosol modules used in this study are
AERO5/6 with ISORROPIA (Nenes et al., 1998, 1999), Modal Aerosol Dynamics for Europe
(MADE) (Ackermann et al., 1998) coupled with SOA scheme based on the Volatility Basis Set
(VBS) approach (SOA_VBS) (Murphy and Pandis, 2009) and Goddard Chemistry Aerosol
Radiation & Transport Model (GOCART) (Chin et al., 2002). The ISORROPIA module has two
versions. The second version (ISORROPIAv2) comes out after CMAQv5.0 with updates in the
thermodynamics of crustal species, the speciation scheme and formation pathway of $SO_4^{2-}$
(Fountoukis and Nenes, 2007). These updates are supposed to lead different model performances
on PM between CMAQv4.7.1 (with first version of ISORROPIA) and CMAQv5.0.2 (with second
version). The GOCART module does not include formations of $NO_3^-$ and $NH_4^+$, therefore it builds
the total PM by combining $SO_4^{2-}$, OC and BC. Please refer to supplementary sect. S1.1 for more
information.



(2)      Emissions/modules of dust. Another important pathway of forming particles is the
disruption or weathering of solid (i.e. soil and rocks) and bursting bubbles of liquid (i.e. sea spray).
The particles formed in this way are generally in coarse mode. Dust emissions have affected large
extension of areas in China. The floating dust of the Takalmakan Desert and Gobi Desert in north-
western China can transport a long distance over the northern China and even reach the Pacific
Ocean (Huang et al., 2008;Iwasaka et al., 2003;Liu et al., 2003;Wang et al., 2018). The
heterogeneous reactions taken place on the surface of dust make it more complicated to simulate
dust in models (Dong, et al., 2016;Dong, et al., 2018;Wang et al., 2017;Wang et al., 2018). Four
models in this study employ dust emissions/modules. All modules adopt parameterization methods
to estimate the floating dust as response to winds (Foroutan et al., 2017;Wang et al., 2012). The
biggest differences lay on the estimation of the dust uplifting processes. Parameters such as dust
source maps and algorithms for friction velocity could result in large differences in model
performances (Ma et al., 2019). M12 and M14 models adopt the same module based on dust
uplifting theory of Gillette and Passi (1988) and modified by Han et al. (2004). M10 model uses
the online generated emission of dust by the GOCART model (Ginoux et al., 2001). M11 model
employs the module of dust with heterogeneous reactions on dust surface (Wang et al., 2017).
Please refer to supplementary sect. S1.2 for more information.
(3)      Removal processes of PM. Wet and dry depositions are the most important pathways to
remove PM from the atmosphere. Wet deposition removes gases and aerosols with rain droplets
and dry deposition is mainly driven by gravitation. The efficiencies of depositions affect the
amounts of aerosols remained in the atmosphere, therefore the removal processes influence the
model accuracies on predicting PM. In this study, all models except M12 use the same dry
deposition scheme from Wesely (1989). M12 adopts the updated scheme by Zhang et al. (2003)
based on Wesely (1989). Please refer to supplementary sect. S1.3 for more information.
**3 Result and discussion**
**3.1 Brief results of model performance evaluation**
Figure 1 and table 2 show the MMM performances on $PM_{10}$, $PM_{2.5}$ and SNA over EA. This section
summaries the major findings of model evaluation since this article focuses more on model
intercomparison. Please refer to supplementary sect. S2 for detailed evaluation results. Evaluation
of model performance on aerosols can also be found in Chen et al., 2019. In the following content,



the model biases are presented by mean bias (MB) and normalized mean bias (NMB). The inter-
model variations are demonstrated by 1 standard deviation among models (1sd) and 1sd%
(calculated as $100\% \times 1sd/MMM$).
Overall, the MB and NMB of surface $PM_{10}$, $PM_{2.5}$ and SNA are -25 µg m$^{-3}$ (-30%), -7 µg
m$^{-3}$ (-15%). -0.7 µg m$^{-3}$ (-19%), -0.05 µg m$^{-3}$ (-3%) and 0.1 µg m$^{-3}$ (12%), respectively. For central
EA (China), the $PM_{10}$ concentrations in northwest China are largely underestimated by 40 µg m$^{-3}$
(MB) and 300% (NMB). The inter-model variations are high around the Taklamakan Desert
(supplementary fig. S2) and Gobi Desert (supplementary fig. S2) (80-110 µg m$^{-3}$ (1sd) and >210%
(1sd%)), due to the implementation of dust emissions/modules in four models. Underestimation
of $PM_{10}$ concentrations is also found in the Hebei-Beijing-Tianjin (HBT) region (supplementary
fig. S2) in northeast China (-68 µg m$^{-3}$ and -46%). However, the inter-model variations of $PM_{10}$
(20-30 µg m$^{-3}$ and 10-30%) of this region are not as high as the model bias, which indicates a
systematic underestimation of $PM_{10}$ by models. On the other hand, the model bias of $PM_{2.5}$ (-8 µg
m$^{-3}$ and -11%) in this region is much lower than that of $PM_{10}$, which reveals model underestimation
of coarse mode of particles (PMC). The $PM_{10}$ concentrations are also generally under-predicted
by 30-40 µg m$^{-3}$ (50-100%) at the sites near the east coast of China.
For eastern EA (Japan and Korea), the $PM_{10}$ and $PM_{2.5}$ concentrations are underestimated
by 15 µg m$^{-3}$ (52%) and 4 µg m$^{-3}$ (40%), respectively. The $SO_4^{2-}$ concentrations are underestimated
at most sites, with high inter-model variations in Japan (3 µg m$^{-3}$ and 90-100%). The monthly
trends of $SO_4^{2-}$ and $NO_3^-$ are poorly simulated due to the underestimation of $SO_4^{2-}$ concentrations
during January to March and the underestimation of $NO_3^-$ concentrations during May to July
(supplementary fig. S5). For northern EA (Russia and Mongolia), only model performances on
SNA are evaluated due to lack of $PM_{10}$ and $PM_{2.5}$ observations during the research periods. The
model biases for different sites vary largely for $SO_4^{2-}$ (-80% to 36%), $NO_3^-$ (-72% to 237%) and
$NH_4^+$ (-81% to 58%), which indicates high uncertainties in the emission inputs. Localized data are
required to update the current emissions in this region, which is derived from Regional Emission
Inventory in Asia version 2.1 for 2000-2008 (Li et al., 2017) (see more details in supplementary
sect. S2). For southern EA (Cambodia, Lao PDR, Myanmar, Thailand, Vietnam, Indonesia,
Malaysia and Philippines), the $PM_{10}$ concentrations are slightly underestimated by 18 µg m$^{-3}$
(45%), while SNA concentrations are overestimated by 0.5 µg m$^{-3}$ (14%), 0.4 µg m$^{-3}$ (28%) and


1.2 µg m$^{-3}$ (124%) for $SO_4^{2-}$, $NO_3^-$ and $NH_4^+$, respectively. It is hard to give a comprehensive
review on this region due to insufficient observations.

The AOD columns (supplementary figs. S6-S7) in north-western EA (near Taklamakan

desert) and south EA (especially around the Himalayas Mountains (supplementary fig. S2)) are
somewhat underestimated, especially in the spring season, which agrees with the underestimation
of $PM_{10}$ in these regions. On the other hand, the overestimation of AOD column in southeast China
in spring and winter (upmost 0.4) is not in accordance with the good model performances on $PM_{10}$
in this region. This inconsistency may correlate with the large inter-model variations of AOD
column in spring (1sd = 0.7) and winter (1sd = 0.4) in this region.

We also compare the model performances with global-scale model study. The Task Force

on Hemispheric Transport of Air Pollution (TF HTAP) is an inter-comparison study of global and
regional models to assess the impact of hemispheric transport of air pollutants on regional
atmosphere. The second phase of HTAP (HTAP-II) involved more than twenty global models to
simulate the air quality in 2010 (Galmarini et al., 2017). Most models utilize coarse-resolution
grids at about 2°-3°. The HTAP-II and MICS-Asia III share some common points like using the
same emission inventory in East Asia (Li et al., 2017) and using the same observation dataset to
evaluate $PM_{10}$ (more than 100 EANET and API sites) and $PM_{2.5}$ (two EANET sites) (Dong et al.,
2018). The MB of $PM_{10}$ over EA is -30.7 µg cm$^{-3}$ and -11.2 µg cm$^{-3}$ for HTAP-II and this study,
respectively. And the MB of $PM_{2.5}$ is -1.6 µg cm$^{-3}$ and -4.3 µg cm$^{-3}$ for HTAP-II and this study,
respectively. Both studies find underestimation of $PM_{10}$ concentrations, while $PM_{2.5}$
concentrations are well produced. Models of MICS-Asia III perform slightly better than those of
HTAP-II with lower model bias in $PM_{10}$, probably taking the advantage of finer resolutions of
model grids.

The so-call "diagnostic evaluation" approach is adopted to check the model bias oriented

by individual process (Dennis et al., 2010). According to the evaluation above, the following
four processes are identified as the main reasons for the model bias with observation and the
possible reasons for model differences:
(1) Source of PM: sect. 3.2 quantifies the uncertainties brought by model inputs, including the

spatial and vertical allocations of emissions and IC/BC.





(2) Formation of PMF: sect. 3.3 investigates the gas-particle conversion of S and N among
different models and the impacts on model performances.

(3) Formation of PMC: sect. 3.4 assesses the model abilities in reproducing the spatial and
temporal distributions of PM in regions affected by dust storm. A comparison is conducted
between models with and without dust emissions/modules.

(4) Removal of PM from the atmosphere: sect. 3.5 compares the model performances in simulating
the amounts of deposition and the efficiencies of wet and dry depositions.

### 3.2 Model inputs and initial/boundary conditions

The model inputs determine the sources of PM. The anthropogenic emissions (including biomass burning emission), biogenic emissions and volcanic emissions are provided by topic II of MICS-Asia. But some natural emissions such as dust and sea salt emissions are prepared by each modelling group. It is important to quantify the influences brought by model inputs before further comparison. Most models did not submit the simulation emission files, therefore the black carbon and CO concentrations are used as indicators of emissions since they weakly react with other species. The modelled concentrations of black carbon at first layer are shown in supplementary fig. S9. Note that the heights of the first layer are 57 meters for all WRF-CMAQ models and M8, but vary from 29 meters to 100 meters for the others (Table 1). Most models produce similar domain-average concentrations of black carbon (ranging from 0.33-0.44) except M5 (0.18) and M10 (0.58). For the six models with 57 meters as the height of the surface layer, the largest difference is about 0.25 $\mu g\ m^{-3}$ (70%). The spatial distributions of black carbon are highly consistent among models. We plot the domain-average CO concentrations at each vertical layer for models (since models do not provide layer height) to compare the vertical allocations of emissions among models (supplementary fig. S10). The CO columns (sum of all vertical layers) among models can vary by up to 756 ppb (22%) for the seven models with 40 vertical layers.

     To assess the impacts brought by using different IC/BC, the results of M1 and M2 models are compared since they use the same model configurations except the IC and BC (supplementary fig. S11). M1 uses the downscale results from GEOS-Chem global model while M2 model uses the default values of CMAQ. The difference between two models are upmost ±3 $\mu g\ m^{-3}$ for black carbon, 20-40 $\mu g\ m^{-3}$ for $PM_{10}$ and $PM_{2.5}$ (high in northern Indian and Southeast Asia), -8 $\mu g\ m^{-3}$ for $SO_4^{2-}$, 2-6 $\mu g\ m^{-3}$ for $NO_3^-$ (high in middle China and northern India) and about 2 $\mu g\ m^{-3}$ for





$NH_4^+$ (high in eastern China). Overall, the results from M1 are about 40-50% higher than M2
around the edges of the simulation domain. This agrees with what we have expected since the
inputs from GEOS-Chem include the long-range transport of pollutions from outside of the
simulation domain. On the other hand, the differences in the centre domain are relatively smaller.
M1 model produces 20-30% higher concentrations of $PM_{10}$ and $PM_{2.5}$ in south EA and 10% higher
concentrations of $PM_{10}$, $PM_{2.5}$ and $NO_3^-$ in centre China than M2. The 10-20% negative differences
in $SO_4^{2-}$ and $NH_4^+$ concentrations between M1 and M2 are probably results of changes in chemical
reactions.

The results demonstrate considerable impacts of emission inputs and IC/BC on model

results. In the following analyses, indicators (i.e. sulphur oxidation ratio (SOR)) are used in
addition to direct model outputs (i.e. $SO_4^{2-}$ concentrations) to exclude the influences and focus
more on the differences caused by model mechanisms.
**3.3 Gas-particle conversion**
The following two indicators are calculated to illustrate the gas-particle conversions of S and N.
$$SOR = \frac{n\text{-}SO_4^{2-}}{n\text{-}SO_4^{2-} + n\text{-}SO_2} \qquad (1)$$
$$C(NO_2) = \frac{n\text{-}NO_3^-}{n\text{-}NO_3^- + n\text{-}NO_2} \qquad (2)$$
where $n\text{-}SO_4^{2-}$, $n\text{-}SO_2$, $n\text{-}NO_3^-$ and $n\text{-}NO_2$ are the mole concentrations of $SO_4^{2-}$ particle, $SO_2$ gas,
$NO_3^-$ particle and $NO_2$ gas. The $C(NO_2)$ indicator only has $NO_3^-$ and $NO_2$ in the denominator due
to the limitation of observation data. But it still can portrait the conversion of N between gas phase
and particle phase.
The $SOR$ values (supplementary fig. S12) are lowest around the HBT region in north-eastern China
(10-40%) and highest in south-western China (60-80%). The X-CMAQ models (including WRF-
CMAQ and RAMS-CMAQ) produce similar $SOR$ patterns, except that the CMAQv5.0.2 models
(M1 and M2) predict 10% higher $SOR$ in the HBT region than the CMAQv4.7.1 models (M4, M5
and M6). For the X-Chem models (including WRF-Chem, GEOS-Chem and NHM-Chem), the
two WRF-Chem models (M7 and M8) produce similar magnitudes and distributions of $SOR$ in all
regions, except the south-western China (around Tibet (supplementary fig. S2)) and the open
oceans, while the NHM-Chem (M12) and GEOS-Chem (M13) models produce slightly higher





*SOR* values over the whole simulation domain. The differences between the X-CMAQ and the X-
Chem models are significant over the inland regions of northern and eastern China, Japan and
southern EA, where the X-CMAQ models generally predict 5-20% higher *SOR* than the X-Chem
models. Similarly, the X-CMAQ models generally give 20% higher *C(NO₂)* values (supplementary
fig. S13) than the WRF-Chem models, especially in eastern EA. The *C(NO₂)* of M8 is extremely
low due to unreasonably low $NO_3^-$ concentrations, which is considered as outlier in this study.

Figure 2 shows the gas-particle conversions of S and N by models and observation at the

EANET sites. The red bars represent concentrations of gases and the black bars represent
concentrations of aerosols. The values with blue color above the bars are observed and modelled
*SOR* and *C(NO₂)* values. Results for individual sites are available in supplementary fig. S14.
According to fig. 2(a), the total amount of S ($SO_2$ gas+$SO_4^{2-}$ particle) is about 0.15 μmole(S) m$^{-3}$.
Most models have biases on this value, especially the moderate underestimation by M7, M8 and
M13. On the other hand, the *SOR* value (0.25) is well simulated by M1 (0.26), M2 (0.20), M10
(0.29) and M13 (0.26). Other models generally under-predict the *SOR* value except M12 (0.33)
and M14 (0.57). The WRF-CMAQv5.0.2 models (M1 and M2) produce higher *SOR* than WRF-
CMAQv4.7.1 models (M4, M5 and M6), probably attributed to the updates in the formation
pathway of $SO_4^{2-}$.

Figure 2(b-e) show the results in different regions. In northern EA, the total amount of S is

underestimated by all models except M13 and M14. However, the *SOR* value (0.12) is well
reproduced by most models (0.08-0.20) except M12 (0.25) and M10 (0.32). After checking the
model performances at the five sites in northern EA (supplementary fig. S14 (a-e)), we found that
the $SO_2$ concentrations at three out of the five sites are largely underestimated by most models,
while the $SO_4^{2-}$ concentrations are well simulated. Therefore, the model biases in northern EA sites
could come from insufficient S in emission inputs, which agrees with our finding in the emission
inputs of this region as mentioned in sect. 3.1 and supplementary sect. S2. There is only one site
available for central EA. Most models (except M12 and M13) have largely underestimated the
*SOR* value, while M14 has largely overestimated it. For eastern EA, the total amount of S is well
captured by all models except M11, M12 and M14. The *SOR* value (0.55) is generally
underestimated by all models except M10 (0.55) and M14 (0.71). For southern EA, the total
amount of S is generally overestimated by all models except M13, while the *SOR* value is



underestimated by all models except M13 and M14. Overall, the models have both positive and
negative biases in simulating the total amounts of S, but generally underestimated the *SOR* values
in all regions. Furthermore, the modelled *SOR* values vary largely among models (ranging from
0.12 to 0.57), resulting in a large inter-model difference (1sd% = 50%). This variation is of the
same magnitude as the variation of $SO_4^{2-}$ concentration (1sd% = 50%). The results suggest that
differences in gas-particle conversion among models could account largely for the models'
inconsistency in simulating the $SO_4^{2-}$ concentrations.

Figure 2(f-h) compares the gas-particle conversion of N with the *C(NO₂)* indicator. Only

one site in China and one site in Japan have both $NO_2$ and $NO_3^-$ observations. At the Hongwen
sites in China, all models except M5 underestimate the sum of $NO_2$ and $NO_3^-$, but the modelled
*C(NO₂)* values are close to the observation (0.18) except M5 (0.07), M8 (0.00) and M12 (0.40).
Similar to the results of S conversion, the newer version of WRF-CMAQ model generally produces
higher *C(NO₂)* than the older version, but the differences between the two are smaller. At the
Banryu site in Japan, the sum of $NO_2$ and $NO_3^-$ is well simulated by all models except M8. The
*C(NO₂)* (0.19) value is also well simulated by all models except M8 (0.00), M12 (0.53) and M14
(0.77). Overall, the model accuracy on *C(NO₂)* is slightly higher than that on *SOR* according to the
comparison with observed values. Models also have higher consistencies on *C(NO₂)* than *SOR*
(also shown in supplementary figs. S12-S13). However, further validation is required due to the
limited number of observations for the conversion of N.

### 3.4 Implementation of dust emissions/modules in models

The PMC concentrations at surface layer are calculated by subtracting $PM_{2.5}$ from $PM_{10}$
(supplementary fig. S15). Most models show very low ($< 2\mu g\ m^{-3}$) concentrations of PMC around
the Takalmakan Desert and the Gobi Desert in northern China except M10, M11 and M14.
According to table 1, these three models use dust emissions/modules in simulations (M12 also
includes dust emissions, but its $PM_{10}$ concentrations over northern China are much lower than the
three models). However, the predicted PMC concentrations for the three models largely differ. The
domain-average concentrations of PMC are 21, 7 and 12 $\mu g\ m^{-3}$ for M10, M11 and M14,
respectively. The distributions of PMC also differ largely over north-west China, where the
impacts of dust are most significant. Different PMC concentrations are also found over oceans,
mainly attributed to the sea-salt emissions in this study. The sea-salt emissions are parameterized



in the models with various formula. In this study, the WRF-Chem models (M7 and M8) do not
account for sea-salt emissions, thus their PMC concentrations over the oceans and seas are not
defined. The two WRF-CMAQ models use the in-line sea-salt emission module of Gong (2003)
and updated by Kelly et al. (2010). They predict consistent distributions of PMC over oceans. M10
and M11 use the same module as the CMAQ models (Gong, 2003), but produce higher PMC on
oceans. M12 adopts the method of breaking wave over seashore by Clarke et al. (2006) and
produces the highest PMC over oceans among all models. Detailed description of the sea-salt
modules can be found in supplementary sect. S1.2.

The implementation of dust emission is expected to improve the model performances, but

how significant could the improvement be? And can models predict the PM concentrations
perfectly at regions affected by dust with current dust emissions/modules? To answer these
questions, all sites are grouped to dust and non-dust sites according to their locations. The sites
located in regions that have been reported to receive severe impacts and rapid deposition of dust
are marked as dust sites (Wang et al., 2004;Wang et al., 2005;Shao and Dong, 2006) (grey-color
shaded areas in supplementary fig. S2). Figure 3(a-b) and table 3 compare the model performances
at the dust and non-dust sites. For the non-dust sites (fig. 3(b)), most models have well captured
the magnitudes of $PM_{10}$ at the "API non-coastal, non-dust" sites (MB = -8% and NMB = -8%).
The sites marked as "API coastal" sites, which are located close to the coastal regions, are all
slightly underestimated by about 25 µg cm$^{-3}$ (30%). Similarly, the PRD and Taiwan sites, which
are also located near the coastal regions, are all underestimated by about 20 µg cm$^{-3}$ (37%). Bias
in sea-salt emissions is the possible reason. Sea-salt emission is reported to contribute to 20-40%
of SNA and $PM_{10}$ over coastal regions (Liu et al., 2015). Including the sea-salt emission in model
simulation can improve the model accuracy with 8-20% increase in $PM_{10}$, SNA, $Na^+$ and $Cl^-$ (Kelly
et al., 2010;Im, 2013). The influence of sea-salt emission is not the focus of this study, but further
study is strongly recommended.

For the dust sites (fig. 3(a)), most models have generally underestimated the $PM_{10}$

concentrations by 10-40 µg cm$^{-3}$ (15-50%). And the three models with dust module perform better
than the others at the dust sites, especially A2, A30, A68, A69, R5 and R18. However, they miss
the high $PM_{10}$ concentrations at sites like R1-R3 and R11, and overestimate the $PM_{10}$
concentrations at sites such as A60 and A80. This indicates that the dust emissions/modules





involved in this study can't fully capture the magnitudes and distributions of dust pollutions over
EA. In addition, the modelled PMC differ a lot with different dust emissions/modules
(supplementary fig. S15). M10 model produces very high PMC over the whole eastern China,
while M11 model only predicts high PMC around the HBT region. Overall, the model performance
on PM over dust regions can be improved largely by including dust emissions/modules. However,
the concentrations and distributions are not yet well captured and large inconsistencies are found
among different dust emissions/modules.

Figure 3(c-d) compares the modelled monthly trends of $PM_{10}$ with observations at the dust

and non-dust sites and figure 3(e) shows the correlations (R) values between models and
observation. For the non-dust sites (Fig. 3(d)), the trends are well caught by most models. The R
values are close to 0.70 for all models except M7 (0.62), M8 (0.58) and M14 (0.63). The WRF-
Chem models (M7 and M8) simulate too low $PM_{10}$ concentrations in winter. M14 model
overestimates the $PM_{10}$ concentrations during March to May. Most models have much lower R
values at the dust sites than the non-dust sites (fig. 3(e)), due to underestimation of the $PM_{10}$
concentrations during winter. For instance, R values of M10 drop from 0.7 at the non-dust sites to
0.11 at the dust sites. Spring (March, April and May) has the largest model biases at the dust sites,
which is coincident with the dust storm season in Asia (Arimoto et al., 2006). M10 and M14
models perform well in most months at both the dust and non-dust sites, taking the advantage of
their dust emissions/modules. But their R values at the dust sites are very low. Future study is
strongly suggested on a better understanding of the seasonal variations of dust pollutions.
**3.5 Wet and dry depositions**
This section compares the main removal processes of PM in the models: wet and dry depositions.
Only M2, M4, M6, M11 and M12 have submitted the main components of S and N depositions,
therefore the following analysis are based on these five models. The model performances on wet
deposition are evaluated with observation data from EANET. Please refer to supplementary sect.
S2.5, table S1 and fig. S8 for details. Overall, wet $SO_4^{2-}$ deposition is generally well simulated by
MMM with NMB of -9%. Wet $NO_3^-$ deposition is underestimated by 29%, due to the large under-
prediction in southern EA. Wet $NH_4^+$ deposition is also underestimated by 40%, especially at the
sites in China, Thailand and Philippine. Large inter-model disagreements are found in simulating
the wet deposition of $SO_4^{2-}$ and $NO_3^-$ at the sites in eastern EA (JP and KR), where the WRF-





CMAQ models (M2, M4 and M6) underestimate the deposition and M11 and M12 models
overestimate the deposition. Models also have large disagreements in simulating wet $NH_4^+$
deposition in southern EA. Dry deposition is not evaluated in this study due to lack of observation
(measurement data are available after 2013).

The total S deposition includes wet depositions of $SO_2$, $H_2SO_4$ and $SO_4^{2-}$ and dry
depositions of $SO_2$, $H_2SO_4$ and $SO_4^{2-}$. The total N deposition includes wet depositions of $NO_3^-$,
$NH_4^+$, $HNO_3$, $NH_3$ and dry depositions of NO, $NO_2$, $NO_3^-$, $NH_4^+$, $HNO_3$ and $NH_3$. Table 4 lists the
domain-total annual-accumulated amounts of S and N depositions by models. The total amounts
of wet S deposition ($D_{Swet}$) range from 10.5 to 31.3 Tg(S) yr$^{-1}$ among models (1sd%=75%). The
estimation by M11 model is two folds higher than the other four models. The inter-model
difference is significant even among the same-type of models with different versions. The
CMAQv4.7.1 models (M4 and M6) produce 12.5 Tg(S) yr$^{-1}$ (M4) and 13.8 Tg(S) yr$^{-1}$ (M6) of
$D_{Swet}$, while the prediction by CMAQv5.0.2 model (M2) is 25% lower. Despite the large
discrepancies in the total amount, all five models agree that over 95% of wet S deposition is wet
$SO_4^{2-}$ deposition. The total amounts of S dry deposition ($D_{Sdry}$) range from 4.3 to 10.6 Tg(S) yr$^{-1}$
among models (1sd% =39%). M11 predicts higher $D_{Sdry}$ than other models and the CMAQv5.0.2
model (M2) predicts 45% lower $D_{Sdry}$ than the two CMAQv4.7.1 models (M4 and M6). Similar to
$D_{Swet}$, all models have high agreements on the proportions of the components.

The total amounts of N wet deposition ($D_{Nwet}$) range from 12.2 to 20.0 Tg(N) yr$^{-1}$ among
models (1sd%=21%). The CMAQ models (M2, M4 and M6) simulate close results (12-15 Tg(N)
yr$^{-1}$), while M11 (20.0 Tg(N) yr$^{-1}$) and M12 (16.5 Tg(N) yr$^{-1}$) simulate slightly higher amounts. As
for the proportion of components, M2, M4, M6 and M12 models predict high proportions of wet
$NO_3^-$ and wet $NH_4^+$ depositions (particle phase), while M11 model produces higher percentages of
wet $HNO_3$ and wet $NH_3$ depositions (gas phase). The total amounts of dry N deposition ($D_{Ndry}$)
range from 3.9 to 14.1 Tg(N) yr$^{-1}$ (1sd%=38%). M12 gives a considerably lower amount than the
other models. Models are quite consistent on the proportions of components.

The modelled deposition is affected by the emission inputs as mentioned in sect. 3.2.
Therefore, two indicators are adopted to exclude the influences: washout ratio of wet deposition
($\lambda_{wet}$) and dry deposition velocity ($V_d$) as calculated by Eqs. 3-4.



$$\lambda_{wet} = \frac{C_{depo}}{C_{surface\_air}} \times 100\% \qquad (3)$$

$$V_d = -F_c / C_{surface\_air} \qquad (4)$$

where $\lambda_{wet}$ is the washout ratio for wet deposition, $C_{depo}$ is the concentration of particles in
deposition and $C_{surface\_air}$ is the concentration of particles at near surface atmosphere. $F_c$ is the
vertical flux of dry deposition and $V_d$ is the deposition velocity. The negative mark indicates the
direction of the dry deposition velocity. $V_d$ is determined by the resistances of air layers. Please
refer to supplementary sect. S1.3 for more information.
Figure 4(a-e) show $\lambda_{wet}$ of S deposition ($\lambda S_{wet}$) by models. The CMAQ models (M2, M4 and
M6) have similar patterns in $\lambda S_{wet}$ over the inland regions, while M12 model predicts 30-90% lower
ratios in India. M11 model generally predicts about 20-70% lower $\lambda S_{wet}$ than the other four models
except India, where the difference could reach upmost 170%. For $\lambda_{wet}$ of N deposition ($\lambda_{Nwet}$) (fig.
4(f-j)), the CMAQv4.7.1 models (M4 and M6) and M12 perform similarly, but the CMAQv5.0.2
model (M2) predicts 30% lower $\lambda_{Nwet}$ in India, Japan and Korea. M11 generally predicts lower
ratios in India (60% lower), Indonesia and Philippines (120% lower) than the CMAQ models.
Figure 5 shows the spatial distributions of $V_d$. For $V_d$ of S deposition ($V_{Sd}$) (fig. 5(a-e)), the CMAQ
models (M2, M4 and M6) simulate very similar spatial distributions. M11 and M12 models predict
0.5 cm s$^{-1}$ lower $V_{Sd}$ than the CMAQ models over the whole inland regions, especially in east China
and India peninsular. For $V_d$ of N deposition ($V_{Nd}$) (fig. 5(f-j)), the CMAQ models (M2, M4 and
M6) predict very similar distributions. M11 and M12 predict about 0.3 cm s$^{-1}$ and 1-2 cm s$^{-1}$ lower
$V_{Nd}$ than the CMAQ models over the inland regions. Overall, large inter-model differences are
found in predicting both the amounts of depositions and the efficiencies of depositions.
**4 Conclusion**
The topic I of the MICS-Asia III aims at (i) evaluating the strengths and weaknesses of current
multiscale air quality models in simulating concentration and deposition fields over East Asia and
(ii) providing suggestions for future model developments. This study compares the performances
of twelve regional models for the prediction of PM concentrations over EA. The participating
models includes WRF-CMAQ (v4.7.1 and v5.0.2), WRF-Chem (v3.6.1 and v3.7.1), GEOS-Chem,
NHM-Chem, NAQPMS and NU-WRF. Evaluation of model performances shows that the mean
biases of MMM are -25 µg m$^{-3}$ (-30%), -7 µg m$^{-3}$ (-15%). -0.7 µg m$^{-3}$ (-19%), -0.05 µg m$^{-3}$ (-3%)





and 0.1 µg m$^{-3}$ (12%) for surface $PM_{10}$, $PM_{2.5}$, $SO_4^{2-}$, $NO_3^-$ and $NH_4^+$ concentrations, respectively.
Four processes/mechanisms are investigated to identify the model biases with observation and the
causes of inter-model differences:
(1) For the sources of PM, we assess the influences of unprescribed natural emissions (i.e. dust
and sea-salt emissions), IC and BC on model performances. The inter-model differences in
surface domain-average black carbon can reach upmost 0.25 µg m$^{-3}$ (70%) and those in
domain-average CO column is about 756 ppb (22%). Using different IC/BC causes about 10-
20% differences in the center of the simulation domain and upmost 40-50% differences at the
edges of the simulation domain for the concentrations of PM and components (based on
comparison between two models). Indicators such as *SOR* are recommended for model
intercomparison to exclude the influences of inconsistent model inputs and IC/BC.
(2) For the formations of PMF, *SOR* and *C(NO$_2$)* values are used to demonstrate the inter-model
differences in gas-particle conversions. The *SOR* values are generally underestimated by most
models at the EANET sites. A generally trend is found that the WRF-CMAQv5.0.2 models
produce the highest *SOR* values among all models, followed by the WRF-CMAQv4.7.1 models
(10% lower in HBT region), the WRF-Chem models and other models (5-20% lower over
inland regions). The inter-model variation on *SOR* (1sd% =50%) is of the same magnitude as
that on $SO_4^{2-}$ concentration. Similar results are found in *C(NO$_2$)*, but models have higher
agreements on *C(NO$_2$)* than *SOR*. The different treatments of gas-particle conversions account
largely for the different model performances on PMF.
(3) For the formations of PMC, the models without dust emissions/modules generate very low
(<2µg m$^{-3}$) PMC concentrations. They can well capture the $PM_{10}$ concentrations at non-dust-
affected sites but underestimate the $PM_{10}$ concentrations at sites affected by dust storms by
upmost 50%. This underestimation is largely improved by implementing dust
emissions/modules (bias reduced to around -20%). However, both the magnitudes and
distributions of dust pollutions are not fully captured. In addition, models employing different
dust emissions/modules show large disagreements on the distributions of PMC.
(4) For the removal of PM from the atmosphere, the amounts of atmospheric deposition vary
largely among models (1sd%) by 75%, 39%, 21% and 38% for $D_{Swet}$, $D_{Sdry}$, $D_{Nwet}$ and $D_{Ndry}$,
respectively. The $\lambda_{wet}$ and $V_d$ indicators are used to exclude the influences brought by model
inputs. For $\lambda_{wet}$, models agree more on the $D_{Swet}$ than $D_{Nwet}$. The largest model inconsistencies





are found in India (upmost 170%), Indonesia and Philippines (upmost 120%). For $V_d$, models
differ more on $D_{Ndry}$ than $D_{Sdry}$, which is opposite to $\lambda_{wet}$. The inter-model differences are
widely found over the inland regions for $D_{Sdry}$ (about 0.5 cm s$^{-1}$) and $D_{Ndry}$ (0.3-2 cm s$^{-1}$).
This paper aims at investigating the potential reasons for model differences on simulating PM$_{10}$
over EA. The main contributions can be concluded as: (1) providing a comprehensive view on the
total budget of S and N aerosols, by including the analysis on model inputs, atmospheric
conversion processes and removal processes. It turns out that the aerosol removal processes can
bring significant uncertainties to inter-model differences; (2) comparing the conversions of S and
N between gas and particle phases among different models as well as with observations. The
comparison with observation makes it possible to both quantify the inter-model differences and
tell which module might be more reasonable. The results can provide important information to
both the model developers and model users; (3) giving an ensemble view on the new updates on
dust modules/emission. Several new updates on dust modules have been published in recent
literature, but there is limited study on the inter-comparison. However, except the processes
mentioned in this study, other factors such as vertical diffusion can also contribute to model
differences. Meanwhile, this study focuses on comparing the model abilities in simulating PM in
2010. The chemical regimes may have changed drastically due the rapid changes of emissions and
implementation of control policies in Asia. Studies on more recent years and heavily polluted
episodes are under preparation.

*Author Contributions*. JT and JSf designed the study. JT processed and analysed the data. JSF,
GRC, SI and ZT contributed to the results and discussions. JSF, ZT, KH, SI, KY, TN, YM, XW,
YL, HJL, JEK, CYL, BG, MK, JZ, MZ, LH and ZW provided modelling data. All co-authors
provided comments to the manuscript.

*Data Availability*. The observation data are introduced with details in supplementary sect. S2.1
with web links of public available datasets. The model data are available upon request.

*Competing interests*. The authors declare that they have no conflict of interest.



*Acknowledgements.* We thank all participating modeling groups of MICS-Asia III. We
acknowledge the support by the Advanced Computing Facility (ACF), Joint Institute for
Computational Sciences of the University of Tennessee. We also thank the support by the Oak
Ridge National Laboratory for computational resources, which is supported by the Office of
Science of the U.S. Department of Energy (contract DE-AC05-00OR22725).

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

# Figures and tables


**Figure 1**

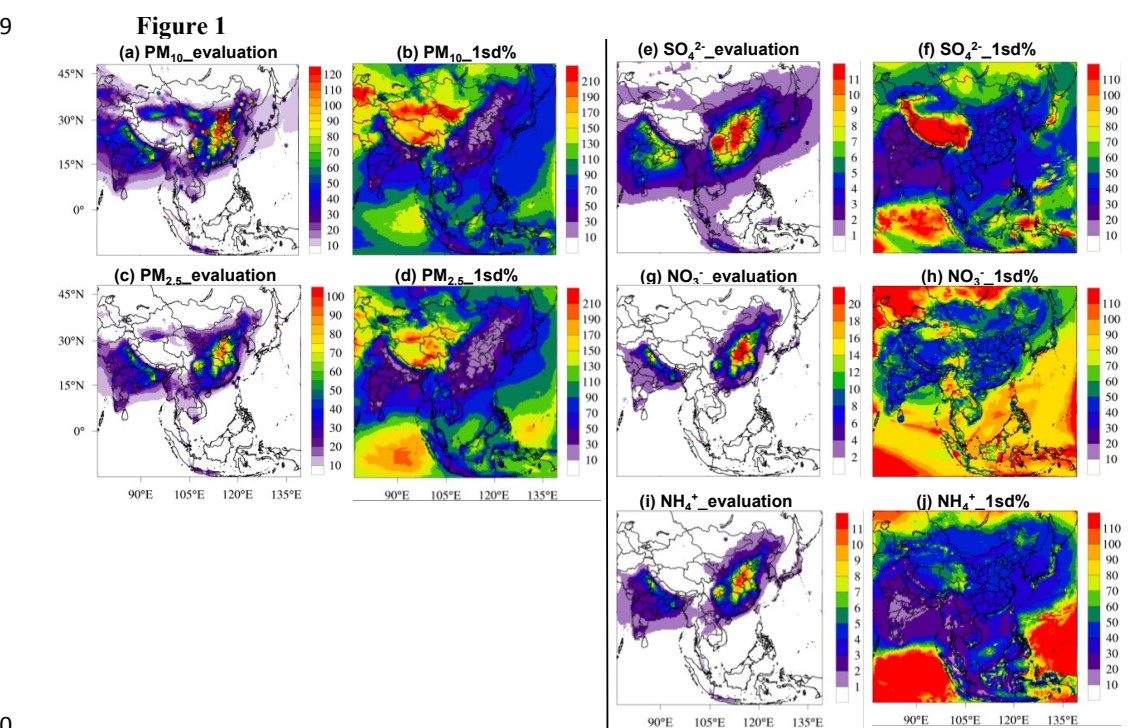

Figure 1 (a,c,e,g,i) Comparison of annual average concentrations of PM and components between MMM (contour)
and observation (markers). The unit is µg m$^{-3}$. (b,d,f,h,j) The inter-model variations among models for PM and
components. 1sd is the standard deviation among models and 1sd% is calculated by dividing 1sd by MMM. The unit
is %.

**Figure 2**



Figure 2 Gas-particle conversions of S and N of observation and models at EANET sites. The unit is μmole (S or N)
m$^{-3}$. The red bars and black bars represent the concentrations of gases and aerosols. The blue-color values above the
bars are observed/modelled *SOR* and *C(NO₂)*. Values are calculated with annual average concentrations. The
concentrations of gases and aerosols are all transferred to μmole (S or N) m$^{-3}$ before calculation. The blue-color
numbers on top-right (e.g. E22) are site numbers. The locations of the sites are illustrated in supplementary fig. S2.
Results for individual sites are shown in supplementary fig. S14.



**Figure 3**

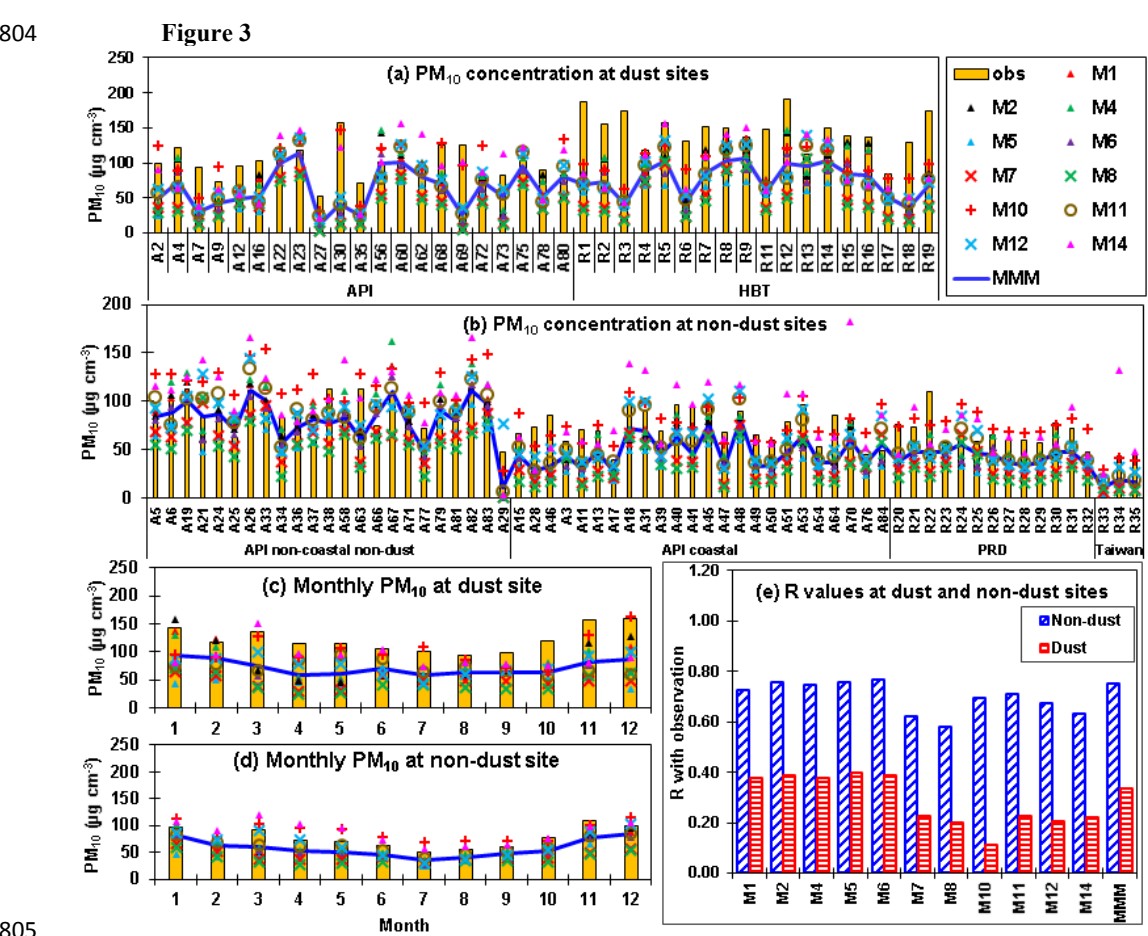

Figure 3 Multi-model performances on (a-b) annual average $PM_{10}$ concentrations at the dust sites and non-dust sites
and (c-d) monthly average $PM_{10}$ concentrations at the dust sites and non-dust sites. X axis for (a-b) indicates site
numbers. The locations of the sites are illustrated in supplementary fig. S2. The yellow bars are observations, the
blue lines are the MMM and different markers represent individual model results. (e) R values of models with
observations at the dust and non-dust sites.

**Figure 4**

(a) M2_S washout %    (b) M4_S washout %    (c) M6_S washout %

(d) M11_S washout %   (e) M12_S washout %

(f) M2_N washout %    (g) M4_N washout %    (h) M6_N washout %

(i) M11_N washout %   (j) M12_N washout %

Figure 4 Washout ratios ($\lambda_{wet}$) of (a-e) S deposition and (f-j) N deposition of models. Values are calculated with
annual accumulated depositions. The unit is %.

**Figure 5**
Figure 5 Dry deposition velocities ($V_d$) of (a-e) S deposition and (f-j) N deposition of models. Values are calculated
with annual accumulated depositions. The unit is cm s$^{-1}$.





**Table 1**
Table 1 Summary of the set-ups of participating models

| | Model | M1 | M2 | M4 | M5 | M6 | M14 |
|---|---|---|---|---|---|---|---|
| | Model Version | WRF-CMAQv5.0.2 | WRF-CMAQv5.0.2 | WRF-CMAQv4.7.1 | WRF-CMAQv4.7.1 | WRF-CMAQv4.7.1 | RAMS-CMAQ |
| Gas | Gas chemistry | SAPRC99 | SAPRC99 | SAPRC99 | SAPRC99 | SAPRC99 | SAPRC99 |
| | Number of species | 77 | 77 | 77 | 77 | 77 | - |
| | Number of reactions | 226 | 226 | 226 | 226 | 226 | - |
| Aerosol | Aerosol chemistry (inorganic) | Aero6 ISORROPIA(v2.0) | Aero6 ISORROPIA(v2.0) | AEO5 ISORROPIA(v1.7) | AEO5 ISORROPIA(v1.7) | AEO5 ISORROPIA(v1.7) | AERO5 ISORROPIA (v1.7) |
| | Aerosol chemistry (organic) | Same as inorganic | Same as inorganic | updated SOA yield parameterization/Carlton et al (2010) | Same as inorganic | Same as inorganic | Same as inorganic |
| | Cloud & Aqueous-phase chemistry | acm_ae6 | acm_ae6 | acm_ae5 | acm_ae5 | acm_ae5 | acm_ae5 |
| | Dry deposition | Wesely (1989) | Wesely (1989) | Wesely (1989) | Wesely (1989) | Wesely (1989) | Aero_depv2 |
| | Wet scavenging | Henry's law | Henry's law | Henry's law | Henry's law | Henry's law | Chang et al (1987) |
| Physical | Horizontal advection | Yamo | Yamo | Yamo | Yamo | Yamo | Hyamo |
| | Vertical advection | PPM | PPM | PPM | PPM | Yamo | Vyamo |
| | Horizontal diffusion | multiscale | multiscale | multiscale | multiscale | multiscale | multiscale |
| | Vertical diffusion | ACM2 | ACM2 | ACM2 | ACM2 | ACM2 | eddy |
| Meteorology | | IAP[1] | IAP[1] | IAP[1] | IAP[1] | IAP[1] | RAMS/NCEP[2] |
| Model set-up | Two way | No | No | No | No | No | No |
| | ICBC | GEOS-Chem | CMAQ default | CHASER | CHASER | CHASER | GEOS-Chem |
| | Vertical layers | 40 | 40 | 40 | 40 | 40 | 15 |
| | Height of 1st layer | 57 m | 57 m | 57 m | 57 m | 57 m | 100 m |
| | Height of top layer | 19,963 m | 19,963 m | 19,963 m | 19,963 m | 19,963 m | - |
| | Grid Size | 45 km | 45 km | 45 km | 45 km | 45 km | 45 km |
| | Dust emission | No | No | No | No | No | Yes |
| | Sea-salt emission | Yes | Yes | Yes | Yes | Yes | Yes |
| | Additional emissions | No | No | No | No | No | No |





Continue table 1

| Model | | M7 | M8 | M10 | M11 | M12 | M13 |
|---|---|---|---|---|---|---|---|
| Model Version | | WRF-Chem 3.7.1 | WRF-Chem3.6.1 | NU-WRF v7lis7-3-3.5.1 | NAQPMS | NHM-Chem | GEOS-Chem |
| Gas | Gas chemistry | RACM-ESRL with KPP | RACM | RADM2 | CBMZ | SAPRC99 | NOx-Ox-HC chemistry mechanism |
| | Number of species | 84 | 84 | 63 | - | 72 | - |
| | Number of reactions | About 249 | About 249 | 157 | - | 214 | - |
| Aerosol | Aerosol chemistry (inorganic) | MADE/VBS | MADE/VBS | GOCART | ISORROPIAv1.7 | ISORROPIA2/MADMS (Kajino et al., 2011) | ISORROPIAv1.7 |
| | Aerosol chemistry (organic) | Same as inorganic | Same as inorganic | Same as inorganic | Same as inorganic | Edney et al. (2007)/MADMS (Kajino, 2011) | Same as inorganic |
| | Cloud & Aqueous-phase chemistry | CMAQ simplified aqueous chemistry | AQCHEM | None | RADM2 | Walcek and Taylor (1986), Carlton et al. (2007) | - |
| | Dry deposition | Wesely (1989) | Wesely (1989) | Wesely (1989) | Wesely (1989) | Kajino et al. (2012) | Wesely (1989) |
| | Wet scavenging | Henry's law | AQCHEM | Grell | Henry's law | Grid scale (Kajino et al., 2012), sub-grid scale convection and deposition (Pleim and Chang, 1992) | Henry's law |
| Physical | Horizontal advection | WRF | WRF | Monotonic | Walcek and Aleksic (1998) | Walcek and Aleksic (1998) | PPM |
| | Vertical advection | $5^{th}$ order monatomic | $5^{th}$ order monatomic | $3^{rd}$ order | | | |
| | Horizontal diffusion | WRF | WRF | $2^{nd}$ order | K-theory | FTCS, Byun and Schere (2006) | Lin and McElroy (2010) |
| | Vertical diffusion | $3^{rd}$ order monatomic | $3^{rd}$ order monatomic | YSU | | FTCS, Mellor-Yamada-Janjic | |
| Meteorology | | WRF/NCEP[2] | WRF/NCEP[2] | WRF/MERRA2[2] | IAP[1] | IAP[1] | GEOS-5[2] |
| Model set-up | Two way | Yes | Yes | No | No | No | No |
| | ICBC | Default | CHASER | MOZART+GOCART | CHASER | CHASER | - |
| | Vertical layers | 40 | 40 | 60 | 20 | 40 | 47 |
| | Height of 1st layer | 29 m | 57 m | 44 m | 48 m | 27 m | - |
| | Height of top layer | 19,857 m | - | 26,168 m | - | - | - |
| | Grid Size | 45 km | 45 km | 45 km | 45 km | 45 km | 0.5° × 0.667° |
| | Dust emission | No | No | Yes | Yes | Yes | - |
| | Sea-salt emission | No | No | Yes | - | - | - |
| | Additional emissions | No | No | Online biogenic + fire | - | - | - |

Note:
1. Models use meteorology inputs provided by the Institute of Atmospheric Physics (IAP), China.
2. Models use own simulated meteorology fields, but applied the same model set-ups as suggested by
IAP.
3. References in the table
Byun, D., and Schere, K. L.: Review of the governing equations, computational algorithms, and other components
of the models-3 Community Multiscale Air Quality (CMAQ) modeling system, Appl Mech Rev, 59, 51-77,
2006.

Carlton, A. G., Turpin, B. J., Altieri, K. E., Seitzinger, S., Reff, A., Lim, H.-J., and Ervens, B.: Atmospheric oxalic
acid and SOA production from glyoxal: Results of aqueous photooxidation experiment, Atmos. Environ., 41,
7588–7602, 2007
Chang, J. S., Brost, R. A., Isaksen, I. S. A., Madronich, S., Middleton, P., Stockwell, W. R., and Walcek, C. J.: A 3-
Dimensional Eulerian Acid Deposition Model - Physical Concepts and Formulation, J Geophys Res-Atmos, 92,
14681-14700, 1987.

Edney, E. O., Kleindienst, T. E., Lewandowski, M., and Offenberg, J. H.: Updated SOA chemical mechanism for
the Community Multiscale Air Quality model, EPA 600/X-07/025, US Environ. Prot. Agency, Durham, NC,
2007.


Kajino, M.: MADMS: Modal Aerosol Dynamics model for multiple Modes and fractal Shapes in the free-molecular
and near-continuum regimes, Journal of Aerosol Science, 42, 224-248, 2011.
Kajino, M., Deushi, M., Maki, T., Oshima, N., Inomata, Y., Sato, K., Ohizumi, T., and Ueda, H.: Modeling wet
deposition and concentration of inorganics over Northeast Asia with MRI-PM/c, Geoscientific Model
Development, 5, 1363-1375, 2012.
Lin, J. T., and McElroy, M. B.: Detection from space of a reduction in anthropogenic emissions of nitrogen oxides
during the Chinese economic downturn, Atmospheric Chemistry and Physics, 11, 8171-8188, 2011.
Pleim, J. E. and Chang, J. S.: A non-local closure model for vertical mixing in the convective boundary layer,
Atmos. Environ., 26A, 965–981, 1992.
Walcek, C. J., and Taylor, G. R.: A Theoretical Method for Computing Vertical Distributions of Acidity and Sulfate
Production within Cumulus Clouds, Journal of the Atmospheric Sciences, 43, 339-355, 1986.
Walcek, C. J., and Aleksic, N. M.: A simple but accurate mass conservative, peak-preserving, mixing ratio bounded
advection algorithm with Fortran code, Atmospheric Environment, 32, 3863-3880, 1998.
Wesely, M. L.: Parameterization of Surface Resistances to Gaseous Dry Deposition in Regional-Scale Numerical-
Models, Atmospheric Environment, 23, 1293-1304, 1989.





**Table 2**
Table 2 MMM performances on annual average PM and components at surface layer (unit: µg m$^{-3}$).

| Data source | PM$_{10}$ | | | | | | | | PM$_{2.5}$ | | | | |
| | All sites | Central EA[1] | | | | Eastern EA[1] | Southern EA[1] | All sites | Central EA[1] | | | Eastern EA[1] |
| | | EANET[2] | API[2] | Ref[2] (HBT) | Ref[2] (PRD) | Ref[2] (TW) | EANET[2] | EANET[2] | | Ref[2] (HBT) | Ref[2] (PRD) | Ref[2] (TW) | EANET[2] |
|---|---|---|---|---|---|---|---|---|---|---|---|---|---|
| Mean Obs | 83.8 | 53.8 | 85.8 | 145.8 | 67.6 | 38.1 | 28.1 | 40.2 | 45.2 | 70.2 | 40.5 | 17.5 | 10.9 |
| Mean MMM | 58.6 | 61.8 | 65.8 | 78.3 | 44.4 | 45.5 | 13.5 | 22.2 | 38.4 | 62.6 | 33.7 | 10.2 | 6.6 |
| $S$[3] | 0.5 | -2.5 | 0.6 | 0.1 | 0.2 | 0.4 | 0.4 | 0.3 | 0.8 | 0.2 | 0.7 | 0.5 | 0.3 |
| $MB$[3] | -25.3 | 8.0 | -20.1 | -67.6 | -23.2 | 7.4 | -14.6 | -18.0 | -6.8 | -7.6 | -6.8 | -7.2 | -4.3 |
| $R$[3] | 0.6 | -0.6 | 0.4 | 0.1 | 0.3 | 1.0 | 0.9 | 1.0 | 0.8 | 0.3 | 1.0 | 1.0 | 0.4 |
| $F$[3] | 74.6 | 66.7 | 80.9 | 61.1 | 92.3 | 66.7 | 46.2 | 100.0 | 82.4 | 75.0 | 100.0 | 100.0 | 66.7 |
| $NMB$[3] (%) | -30.1 | 14.9 | -23.4 | -46.3 | -34.3 | 19.3 | -51.9 | -44.8 | -15.1 | -10.9 | -16.7 | -41.4 | -39.7 |
| $NME$[3] (%) | 34.7 | 50.1 | 29.0 | 46.3 | 34.3 | 19.3 | 51.9 | 44.8 | 26.1 | 25.9 | 16.7 | 41.4 | 39.7 |
| $MFB$[3] (%) | -40.7 | 5.8 | -32.2 | -61.6 | -40.1 | 20.6 | -67.8 | -57.6 | -25.7 | -10.0 | -18.0 | -51.6 | -49.3 |
| $MFE$[3] (%) | 45.3 | 45.2 | 37.9 | 61.6 | 40.1 | 20.6 | 67.8 | 57.6 | 33.0 | 25.5 | 18.0 | 51.6 | 49.3 |
| Number of Sites | 142 | 3 | 89 | 18 | 13 | 3 | 13 | 3 | 17 | 8 | 3 | 3 | 3 |

Note:
1.  Definition of regions: northern EA (Russia and Mongolia), central EA (China), western EA (Japan and Korea)
and southern EA (Cambodia, Lao PDR, Myanmar, Thailand, Vietnam, Indonesia, Malaysia and Philippines).

2.  Monitoring networks: Acid Deposition Monitoring Network in East Asia (EANET) (http://www.eanet.asia/, last
access: 28 May 2018), Air Pollution Indices (API) and Reference dataset provided by the Institute of
Atmospheric Physics Chinese Academy of Science (Ref). Please refer to supplementary S2.1 for detailed
information.

3.  Statistical metrics calculated as following Eqs. 1-5:

$MB \text{ (mean bias)} = \frac{1}{n}\sum_{i=1}^{n}(M_i - O_i)$             (1)

$NMB \text{ (normalized mean bias)} = \frac{\sum_{i=1}^{n}(M_i - O_i)}{\sum_{i=1}^{n}O_i} \times 100\%$     (2)

$NME \text{ (normalized mean error)} = \frac{\sum_{i=1}^{n}|M_i - O_i|}{\sum_{i=1}^{n}O_i} \times 100\%$     (3)

$MFB \text{ (mean fractional bias)} = \frac{1}{n}\sum_{i=1}^{n}\frac{M_i - O_i}{(M_i + O_i)/2} \times 100\%$     (4)

$MFE \text{ (mean fractional gross error)} = \frac{1}{n}\sum_{i=1}^{n}\frac{|M_i - O_i|}{(M_i + O_i)/2} \times 100\%$     (5)

where $M_i$ is the model result, $O_i$ is the observation and n is the sample size. In addition, we use linear fit slope (S),
correlation coefficient (R) and fraction (of model results) within ± 50% of observation (F) as statistical materics to
enable comparison with other studies.






Continue Table 2

| Data source | SO₄²⁻ | | | | | NO₃⁻ | | | | |
|---|---|---|---|---|---|---|---|---|---|---|
| | All sites | North EA[1] | Centra l EA[1] | East EA[1] | South EA[1] | All sites | North EA[1] | Centra l EA[1] | East EA[1] | South EA[1] |
| | | | EANET[2] | | | | | EANET[2] | | |
| Mean Obs | 3.6 | 2.5 | 14.1 | 3.4 | 3.4 | 1.6 | 0.6 | 11.7 | 1.1 | 1.6 |
| Mean MMM | 3.0 | 1.3 | 6.0 | 2.7 | 3.9 | 1.5 | 0.8 | 4.3 | 1.4 | 2.0 |
| $S^3$ | 0.3 | 0.2 | - | 0.3 | 0.3 | 0.4 | 0.5 | - | 0.9 | 0.4 |
| $MB^3$ | -0.7 | -1.2 | -8.1 | -0.7 | 0.5 | -0.05 | 0.1 | -7.3 | 0.2 | 0.4 |
| $R^3$ | 0.6 | 0.5 | - | 0.6 | 0.6 | 0.7 | 0.2 | - | 0.8 | 0.6 |
| $F^3$ | 73.1 | 80.0 | - | 90.9 | 55.6 | 50.0 | 20.0 | - | 72.7 | 42.9 |
| $NMB^3$ (%) | -18.7 | -46.6 | -57.6 | -21.7 | 14.0 | -3.0 | 16.2 | -62.8 | 21.1 | 27.6 |
| $NME^3$ (%) | 51.3 | 50.9 | 57.6 | 31.5 | 72.7 | 70.9 | 109.6 | 62.8 | 41.5 | 101.4 |
| $MFB^3$ (%) | -5.7 | -39.4 | -80.9 | -19.4 | 38.2 | 23.9 | -3.6 | -91.6 | 24.5 | 59.3 |
| $MFE^3$ (%) | 50.3 | 54.0 | 80.9 | 38.1 | 59.8 | 71.3 | 81.4 | 91.6 | 48.7 | 96.7 |
| Number of Sites | 26 | 5 | 1 | 11 | 9 | 24 | 5 | 1 | 11 | 7 |



Continued Table 2

| Data source | NH₄⁺ | | | | |
|---|---|---|---|---|---|
| | All sites | North EA[1] | Central EA[1] | East EA[1] | South EA[1] |
| | | | EANET[2] | | |
| Mean Obs | 1.1 | 0.8 | 6.7 | 0.7 | 1.0 |
| Mean MMM | 1.2 | 0.5 | 2.5 | 0.8 | 2.2 |
| $S^3$ | 0.3 | 0.04 | - | 1.0 | 0.6 |
| $MB^3$ | 0.1 | -0.3 | -4.2 | 0.1 | 1.2 |
| $R^3$ | 0.5 | 0.1 | - | 0.9 | 0.5 |
| $F^3$ | 63.6 | 50.0 | - | 90.9 | 33.3 |
| $NMB^3$ (%) | 11.9 | -34.9 | -62.2 | 10.9 | 123.5 |
| $NME^3$ (%) | 63.5 | 66.2 | 62.2 | 19.5 | 123.5 |
| $MFB^3$ (%) | 21.3 | -28.1 | -90.2 | 13.4 | 87.3 |
| $MFE^3$ (%) | 49.8 | 63.5 | 90.2 | 20.7 | 87.3 |
| Number of Sites | 22 | 4 | 1 | 11 | 6 |







**Table 3**
Table 3 Multi-model performance on annual average concentrations of $PM_{10}$ at the dust and non-
dust sites (unit: $\mu g\ m^{-3}$)

| Dust site | M1 | M2 | M4 | M5 | M6 | M7 | M8 | M10 | M11 | M12 | M14 | MMM |
|---|---|---|---|---|---|---|---|---|---|---|---|---|
| Mean Obs | | | | | | 120.7 | | | | | | |
| Mean MMM | 77.2 | 82.2 | 81.6 | 51.7 | 65.6 | 47.5 | 44.3 | 102.5 | 73.5 | 77.3 | 92.1 | 69.2 |
| S | 0.4 | 0.4 | 0.4 | 0.3 | 0.3 | 0.2 | 0.2 | 0.1 | 0.2 | 0.2 | 0.3 | 0.3 |
| MB | -43.5 | -38.5 | -39.2 | -69.0 | -55.1 | -73.2 | -76.4 | -18.2 | -47.2 | -43.4 | -28.6 | -51.5 |
| R | 0.4 | 0.4 | 0.4 | 0.4 | 0.4 | 0.2 | 0.2 | 0.1 | 0.2 | 0.2 | 0.2 | 0.3 |
| F | 66.7 | 69.2 | 69.2 | 38.5 | 56.4 | 35.9 | 33.3 | 84.6 | 59.0 | 66.7 | 66.7 | 66.7 |
| NMB (%) | -36.1 | -31.9 | -32.4 | -57.2 | -45.7 | -60.6 | -63.3 | -15.1 | -39.1 | -36.0 | -23.7 | -42.6 |
| NME (%) | 38.3 | 35.4 | 36.4 | 57.2 | 46.2 | 60.6 | 63.3 | 32.8 | 42.3 | 40.5 | 36.1 | 42.7 |
| MFB (%) | -49.4 | -44.6 | -44.6 | -83.4 | -64.1 | -92.9 | -98.8 | -19.3 | -51.8 | -46.8 | -31.7 | -56.9 |
| MFE (%) | 51.8 | 48.3 | 48.7 | 83.4 | 64.7 | 92.9 | 98.8 | 36.1 | 55.3 | 51.7 | 44.5 | 56.9 |
| Number of Sites | | | | | | 39 | | | | | | |


Continue Table 3

| Non-dust site | M1 | M2 | M4 | M5 | M6 | M7 | M8 | M10 | M11 | M12 | M14 | MMM |
|---|---|---|---|---|---|---|---|---|---|---|---|---|
| Mean Obs | | | | | | 77.2 | | | | | | |
| Mean MMM | 58.2 | 58.5 | 66.5 | 45.2 | 55.2 | 44.8 | 39.0 | 90.0 | 64.4 | 66.3 | 89.5 | 57.8 |
| S | 1.0 | 1.1 | 1.2 | 0.8 | 1.0 | 0.7 | 0.6 | 1.0 | 1.0 | 0.9 | 1.1 | 0.9 |
| MB | -19.0 | -18.7 | -10.8 | -32.1 | -22.1 | -32.5 | -38.3 | 12.7 | -12.9 | -10.9 | 12.2 | -19.4 |
| R | 0.7 | 0.8 | 0.7 | 0.8 | 0.8 | 0.6 | 0.6 | 0.7 | 0.7 | 0.7 | 0.6 | 0.8 |
| F | 82.5 | 81.0 | 84.1 | 66.7 | 82.5 | 52.4 | 46.0 | 85.7 | 90.5 | 93.7 | 84.1 | 82.5 |
| NMB (%) | -24.6 | -24.2 | -14.0 | -41.5 | -28.6 | -42.0 | -49.5 | 16.5 | -16.6 | -14.1 | 15.8 | -25.1 |
| NME (%) | 30.7 | 30.7 | 27.3 | 41.5 | 31.4 | 43.9 | 50.7 | 25.7 | 26.3 | 26.1 | 30.8 | 28.0 |
| MFB (%) | -36.8 | -37.5 | -25.1 | -59.2 | -41.8 | -62.0 | -75.0 | 13.1 | -24.9 | -20.3 | 8.3 | -34.6 |
| MFE (%) | 42.0 | 42.8 | 35.3 | 59.2 | 44.4 | 64.0 | 76.1 | 23.4 | 33.5 | 31.3 | 29.1 | 37.5 |
| Number of Sites | | | | | | 63 | | | | | | |





**Table 4**
Table 4 Domain-total annual-accumulated S and N depositions of models (Tg(S or N) yr$^{-1}$).
Empty values mean no model submissions or the values are 0.

| Model | Wet S deposition | | | | Dry S deposition | | | |
|---|---|---|---|---|---|---|---|---|
| | $SO_2$ | $H_2SO_4$ | $SO_4^{2-}$ | Total Wet S | $SO_2$ | $H_2SO_4$ | $SO_4^{2-}$ | Total Dry S |
| M1 | 0.06 | - | - | - | - | - | - | - |
| M2 | 0.04 | - | 10.4 | 10.5 | 3.4 | 0.01 | 0.9 | 4.3 |
| M4 | 0.06 | - | 12.5 | 12.5 | 6.6 | 0.01 | 1.1 | 7.6 |
| M5 | - | - | - | - | - | - | - | - |
| M6 | 0.05 | - | 13.7 | 13.8 | 6.3 | 0.01 | 1.4 | 7.7 |
| M7 | - | - | - | - | - | - | - | - |
| M8 | - | - | - | - | - | - | - | - |
| M10 | - | - | - | - | - | - | - | - |
| M11 | 1.1 | 0.3 | 29.9 | 31.3 | 6.9 | 2.2 | 1.5 | 10.6 |
| M12 | - | - | 16.3 | 16.3 | 3.7 | - | 0.4 | 4.2 |
| M13 | 6.0 | - | - | - | - | - | - | - |
| M14 | 0.02 | - | 6.2 | - | 5.4 | - | 3.2 | - |

Continue Table 4

| Model | Wet N deposition | | | | | Dry N deposition | | | | | | |
|---|---|---|---|---|---|---|---|---|---|---|---|---|
| | $NO_3^-$ | $NH_4^+$ | $HNO_3$ | $NH_3$ | Total Wet N | NO | $NO_2$ | $NO_3^-$ | $NH_4^+$ | $HNO_3$ | $NH_3$ | Total Dry N |
| M1 | - | - | - | - | - | - | - | - | - | 4.3 | 6.9 | - |
| M2 | 4.0 | 8.3 | - | - | 12.2 | 0.03 | 0.4 | 0.6 | 0.6 | 2.0 | 7.5 | 11.0 |
| M4 | 5.4 | 7.4 | - | - | 12.8 | 0.03 | 0.3 | 0.7 | 0.5 | 2.8 | 4.7 | 9.0 |
| M5 | - | - | - | - | - | - | 0.5 | - | - | - | - | - |
| M6 | 5.6 | 9.1 | - | - | 14.6 | 0.02 | 0.3 | 0.8 | 0.7 | 2.9 | 6.5 | 11.1 |
| M7 | - | - | - | - | - | - | - | - | - | - | - | - |
| M8 | - | - | - | - | - | - | - | - | - | - | - | - |
| M10 | - | - | - | - | - | - | - | - | - | - | - | - |
| M11 | 1.5 | 2.8 | 8.1 | 7.6 | 20.0 | - | - | 1.3 | 2.4 | 3.3 | 7.1 | 14.1 |
| M12 | 5.4 | 11.0 | - | - | 16.5 | 0.04 | 0.4 | 0.4 | 0.3 | 0.5 | 2.2 | 3.9 |
| M13 | - | - | 4.1 | - | - | - | - | - | - | 4.5 | 4.6 | - |
| M14 | - | - | - | - | - | - | - | - | - | - | - | - |

