# Peer review of "Why models perform differently on particulate matter over East"

_Atmospheric Chemistry and Physics, 2019_

## Referee Comment (RC1) · Anonymous Referee #2 · 30 Oct 2019

General Summary: This study compares performances of 12 regional models from MICS-Asia III in simulating particulate matter during 2010. Performance is evaluated in terms of PM10, PM2.5, sulphate, nitrate, and ammonium. The mean bias of multi-model mean range from -30% to 12% for different species. The analysis focuses on four processes that are likely contributing to the inter-model differences namely natural emissions, gas to particle conversion of sulphur and oxidized nitrogen, role of dust emissions, and deposition parameterization. I think the subject of this paper is suitable for publication in ACP but I have several concerns about the organization and analysis in the paper. Therefore, I recommend major revisions before the paper can be published in ACP.

[Figure]

Firstly, the supplementary information is overwhelming and contains a lot of the material that can be moved to the main paper. Specifically, I suggest moving the evaluation part to the main paper and several key discussions (e.g., supplementary sect. S2.5, table S1 and fig. S8) related to the process analysis.

Second, I have some concerns using CO and BC as surrogates for anthropogenic emissions (see specific comment on lines 258-259).

Third, I have some concerns regarding evaluation with MODIS AOD (see specific comments on the evaluation part).

In addition, there are many grammatical errors that I believe can be removed upon reading by a native English speaker.

Here are my specific and minor comments:

Abstract: Line 32-34: Why would natural emissions affect BC concentrations and CO loading?

Minor comments: Line 58: Change "pollutions" to "pollution".

Line 61: Change "impact evaluation on" to "impact evaluation of"

Line 86: Change "view on" to "view of"

Lines 88-89: Spell out different acronyms.

Table 1: Do you mean AERO5 instead of AEO5 in 6th row.

Line 154-155: This is not correct. GOCART in WRF-Chem accounts for ammonium mass in calculation of PM2.5 and PM10 by multiplying sulfate concentrations by 1.375 to account for missing ammonium. GOCART also includes fine dust and sea-salt in PM2.5.

Line 161: Change "transport" to "travel".

Line 163: Change "taken" to "take"

Line 176: How about turbulent diffusion?

Line 174-180: Please discuss wet deposition parameterization as well.

Line 184: Change "summaries" to "summarizes"

Figure 1: Except for panel (a), I am unable to see the markers for observations in the maps of PM2.5, sulfate, nitrate, and ammonium. I suggest showing standard deviations in absolute values as well because the current plots give an impression that large variability (more than 80%) exists over the regions of lowest concentrations (e.g., Tibetan Plateau) but this is simply a result of division by a very small number.

Line 195: Can you explain in more detail why dust emissions are so different among different models? Is it because of wind speed or soil moisture or source functions?

Line 215: Remove "slightly".

Figure S6: Why isn't there a complete seasonal cycle of MODIS AOD at many locations?

Figure S7: Do white spaces in the maps correspond to AOD below 0.1 or do they also correspond to missing data? Have the model and MODIS AOD been collocated before comparison? Did you use Level 2 MODIS AOD in this comparison?

Lines 219-225: Again, could you please say more about the inter model differences here? Are differences in AOD controlled by differences in aerosol chemical composition or by differences in aerosol size distributions or assumed aerosol optical properties?

Line 234: At line 190, MB in PM10 for this study is reported as -25 ug/m3 but here it is reported as -11.2 ug/m3. Which one is correct?

Line 240: Change "so-call" to "so-called".

Lines 258-259: It is difficult to understand why it is so hard to get access to the emissions used by different groups. Using model simulated CO and BC as surrogates for

emissions is not a good approach because they are strongly influenced not only by emissions but also by transport, deposition, and chemistry (in case of CO). Use of column CO is especially concerning because CO distribution in the free troposphere is primarily controlled by transport (inflow from domain boundaries in case of regional models) and not by emissions. Similarly, regions downwind of strong emission sources will exhibit disproportionately larger concentrations of BC and CO than the emissions. Therefore, I recommend using actual emission fluxes in your analysis in Section 3.2.

Lines 325-326: If the models were able to capture SO2 concentrations, would they have overestimated SO4 considering that they are simulating SO4 reasonably well even with underestimated SO2 concentrations.

Line 364: Sea-salt emissions are controlled via a namelist option in WRF-Chem. Was there a specific reason for turning off sea-salt emissions in M7 and M8?

Line 390: $\mu$g cm-3 or $\mu$g m-3?

---

## Referee Comment (RC2) · Anonymous Referee #3 · 21 Nov 2019

General Comments: The manuscript examines the spatial and temporal variability in aerosol concentration and composition simulated by 12 regional models. The model predictions are evaluated with measurements from the different monitoring networks. It also quantifies the ensemble mean bias through several processes, namely model inputs, gas to particle conversion, the impact of sea-salt and dust emissions parameterization, as well as deposition processes. The topic and overall approach fit with ACP. However, I feel that there are some aspects need to be discussed in more detail and the paper needs some improvement before being published in ACP.

As Reviewer #2 already mentioned, the supplementary material contains a very large

number of figures that could easily be added in the main text. In the current version, the readers have to go back and forth from the main text to the supplementary material.

It would be nice to have a more in-depth investigation of possible causes that could explain the substantial differences between simulations. As an example, there are large differences between simulations performed using the same model (see Figs.2, S9, S12 and S13). The M4 and M5 models seem to share similar setup and use the same input data, except the organic chemistry treatment. Then, how could the authors explain the different modelled spatial distribution of black carbon, SOR or C(NO2), as well as the different modelled concentrations of NO2 and NO3?

The logical connection between the manuscript and the SM needs to be improved. For example, the discussion associated with Fig. S8 appears after the one associated with Fig. S12.

Specific comments

Lines 104-105: it repeats information already provided in the introduction

Line 115: use "IC/BC" instead "BC"

Lines 164-165: Why the online calculation of natural emissions of dust and sea salt particles fully embedded within the WRF-Chem is not included?

Line 185: move this information into the main text

Lines 185-186: briefly describe the main findings of Chen et al., 2019

Lines 191-193: should be an NMB of -300%?

Lines 196-197: Fig. S3 instead of Fig. S2?

Lines 205-208: For northern and south-eastern EA these findings are not applicable. Could the authors explain why this behaviour is seen only for eastern EA, but not for the other analysed regions?

Lines 258-260: As Reviewer #1 already mentioned, using CO and BC concentration as proxies for emissions won't give a good indication of the emission's spatial distribution.

Lines 266-269: the models indeed do not provide the layer height in meters, but this can be easily calculated

Lines 273-276: these differences represent averaged values over the domain? It is difficult to connect these numbers with Fig S11. What do you mean by 2-6 ug m-3 for NO3? Do these numbers represent the range of differences? From Fig, S11 the range seems to be between -4 and 6 ug m-3.

Lines 277-279: have you compared these values with observations?

Lines 297-300: Why? A different aerosol treatment?

Figs. 2 and S13 and subsequent discussion: Could the authors provide an explanation for the really low NO3 concentration modelled with M8? Have they considered the M8 model when the ensemble mean was compared against observation? Using such outliers, the MMM mean can even be deteriorated compared to individual simulations.

Fig. 2 c) M3 should be M4

Lines 323-330: if the S emissions were insufficient, why the M13 and M14 models provide reasonable results?

Section 3.4: same as before, why not using the online calculation of natural emissions of dust particles within WRF-Chem?

Line 374: "perfectly" is a strong word

Lines 432-433: Table 4 and the associated discussion. Please be consistent when calculating the total wet and dry deposition of S/N and doing subsequent analysis.

Lines 460-473: In the case of wet deposition of S and N, M11 shows a very different spatial pattern compared with the other models. What is the reason for this? It would

be nice if the authors will try to analyse in detail the causes of these differences.

---

## Author Comment (AC1) · 15 Feb 2020

**General Summary:**

This study compares the performances of 12 regional models from MICS-Asia III in simulating particulate matter during 2010. Performance is evaluated in terms of $PM_{10}$, $PM_{2.5}$, sulphate, nitrate, and ammonium. The mean bias of multimodel mean range from -30% to 12% for different species. The analysis focuses on four processes that are likely contributing to the inter-model differences namely natural emissions, gas to particle conversion of sulphur and oxidized nitrogen, the role of dust emissions, and deposition parameterization. I think the subject of this paper is suitable for publication in ACP but I have several concerns about the organization and analysis in the paper. Therefore, I recommend major revisions before the paper can be published in ACP.

Firstly, the supplementary information is overwhelming and contains a lot of the material that can be moved to the main paper. Specifically, I suggest moving the evaluation part to the main paper and several key discussions (e.g., supplementary sect. S2.5, table S1, and fig. S8) related to the process analysis.

Second, I have some concerns using CO and BC as surrogates for anthropogenic emissions (see specific comment on lines 258-259).

Third, I have some concerns regarding evaluation with MODIS AOD (see specific comments on the evaluation part).

In addition, there are many grammatical errors that I believe can be removed upon reading by a native English speaker.

Response: We would like to thank the reviewer for the valuable comments. Following are the responses to the comments.

**Here are my specific and minor comments:**

**2Q1: Comment:** Abstract: Line 32-34: Why would natural emissions affect BC concentrations and CO loading?

Response: Thank you for pointing out the question. All models used the same anthropogenic and natural emissions. Thus, the impact of different natural emissions on model performance is not an application in this study. However, mismatch during the temporal and vertical treatment of emission files by different modelling groups has caused differences in the model inputs (Itahashi et al., 2019). In addition, different model set-up such as the heights of first vertical layer and spatial resolutions of model grids also affect the inter-model comparison on direct model outputs such as surface PM concentrations. Therefore, we decided to use indicators (such as SOR) instead of direct model outputs to facilitate comparison on model mechanisms. For this reason, we removed our discussion on the impacts of emissions and IC/BC on model performance (section 3.2 in the old manuscript) and focused on the comparison of the three processes: gas-aerosol portioning, dust mechanism and wet and dry deposition efficiency.

Reference: Itahashi, S., Ge, B., Sato, K., Fu, J. S., Wang, X., Yamaji, K., Nagashima, T., Li, J., Kajino, M., Liao, H., Zhang, M., Wang, Z., Li, M., Kurokawa, J., Carmichael, G. R., and Wang, Z.: MICS-Asia III: Overview of model inter-comparison and evaluation of acid deposition over Asia, Atmos. Chem. Phys. Discuss., https://doi.org/10.5194/acp-2019-624, in review, 2019.

**2Q2: Comment:** Minor comments: Line 58: Change "pollutions" to "pollution".

Response: We have changed it in the manuscript.

**2Q3: Comment:** Line 61: Change "impact evaluation on" to "impact evaluation of"

Response: We have revised the manuscript.

**2Q4: Comment:** Line 86: Change "view on" to "view of"

Response: We have made the change in the manuscript.

**2Q5: Comment:** Lines 88-89: Spell out different acronyms.

Response: We added the full names of the models in the manuscript.

**2Q6: Comment:** Table 1: Do you mean AERO5 instead of AEO5 in 6th row.

Response: Table 1 has been removed from the manuscript.

**2Q7: Comment:** Line 154-155: This is not correct. GOCART in WRF-Chem accounts for ammonium mass in calculation of $PM_{2.5}$ and $PM_{10}$ by multiplying sulfate concentrations by 1.375 to account for missing ammonium. GOCART also includes fine dust and sea-salt in $PM_{2.5}$.

Response: Thank you for pointing out this problem. We removed the description on the model mechanism as they have already been introduced in detail in our companion paper (Chen et al., 2019). The description of the GOCART model is updated in that paper.

Reference: Chen, L., Gao, Y., Zhang, M., Fu, J. S., Zhu, J., Liao, H., Li, J., Huang, K., Ge, B., Wang, X., Lam, Y. F., Lin, C.-Y., Itahashi, S., Nagashima, T., Kajino, M., Yamaji, K., Wang, Z., and Kurokawa, J.: MICS-Asia III: multi-model comparison and evaluation of aerosol over East Asia, Atmos. Chem. Phys., 19, 11911–11937, https://doi.org/10.5194/acp-19-11911-2019, 2019.

**2Q8: Comment:** Line 161: Change "transport" to "travel".

Response: We have made the change in the manuscript.

**2Q9: Comment:** Line 163: Change "taken" to "take"

Response: We have made the change.

**2Q10: Comment:** Line 176: How about turbulent diffusion?

Response: We have revised the description on the mechanism of dry deposition as follows:

Line 137: "Dry deposition is mainly driven by turbulent and molecular diffusion processes."

**2Q11: Comment:** Line 174-180: Please discuss wet deposition parameterization as well.

Response: We added the following sentences to introduce the wet deposition parameterization:

Lines 124-136: "Wet deposition removes gases and aerosols from the atmosphere by rain droplets, involving both in-cloud scavenging (rainout) and below-cloud scavenging (washout). The gases in the atmosphere are dissolved in the raindrop and then removed from the atmosphere. For the non-reactive gases, the removal rate depends on the solubility of gases and is a function of Henry's Law. Particles participate in the cloud condensation nuclei in the presence of supersaturation water vapor and then grow into cloud droplets. In this study, only M2, M4, M6, M11 and M12 have submitted the main components of S and N depositions. All these models use the same wet deposition scheme based on Henry's law. The efficiency of wet deposition is assessed by the so-called "washout ratio", calculated as the ratio of particle concentrations in the deposition to particle concentrations in surface air as shown in Eq. 1.

$$\lambda_{wet} = \frac{C_{depo}}{C_{surface\_air}} \times 100\% \qquad (1)$$

where $\lambda_{wet}$ is the washout ratio for wet deposition, $C_{depo}$ is the concentration of particles in deposition and $C_{surface\_air}$ is the concentration of particles at near surface atmosphere."

**2Q12: Comment:** Line 184: Change "summaries" to "summarizes"

Response: We have made the change in the manuscript.

**2Q13: Comment:** Figure 1: Except for panel (a), I am unable to see the markers for observations in the maps of PM$_{2.5}$, sulfate, nitrate, and ammonium. I suggest showing standard deviations in absolute values as well because the current plots give an impression that large variability (more than 80%) exists over the regions of lowest concentrations (e.g., Tibetan Plateau) but this is simply a result of division by a very small number.

Response: The model performance on aerosols has been updated and discussed in more detail in our companion paper of Chen et al., 2019. All the discussion and figures/tables related to model evaluation have been removed from the manuscript.

The absolute values for standard deviations are shown in supplementary Fig. S2.

**2Q14: Comment:** Line 195: Can you explain in more detail why dust emissions are so different among different models? Is it because of wind speed or soil moisture or source functions?

Response: We added an explanation to the model differences on dust as follows:

Lines 292-294 "The differences among the models mainly come from the different parameterizations such as source functions, dust-lifting mechanisms and size distributions of particles (Chen et al., 2019)."

**2Q15: Comment:** Line 215: Remove "slightly".

Response: We have removed it from the manuscript.

**2Q16: Comment:** Figure S6: Why isn't there a complete seasonal cycle of MODIS AOD at many locations?

Response: The model performance on AOD has been evaluated in our companion paper (Chen et al., 2019) with all available MODIS data. We removed the model evaluation on AOD from this manuscript.

**2Q17: Comment:** Figure S7: Do white spaces in the maps correspond to AOD below 0.1 or do they also correspond to missing data? Have the model and MODIS AOD been collocated before comparison? Did you use Level 2 MODIS AOD in this comparison?

Response: The model and MODIS AOD haven't collocated before comparison. And Level 2 MODIS AOD is used for comparison. We updated the model performance on AOD in our companion paper (Chen et al., 2019) and removed the related text and figures/tables from this manuscript.

**2Q18: Comment:** Lines 219-225: Again, could you please say more about the inter model differences here? Are differences in AOD controlled by differences in aerosol chemical composition or by differences in aerosol size distributions or assumed aerosol optical properties?

Response: AOD is affected by both the chemical composition and particle size distribution. In addition, different model treatments and set-up (as we mentioned in our answer to Q1) could also affect the model results of AOD. We decided to use indicators instead of direct model output like AOD for inter-model comparison. Thus, the discussion related to AOD has been removed from the manuscript.

**2Q19: Comment:** Line 234: At line 190, MB in $PM_{10}$ for this study is reported as -25 ug/m3 but here it is reported as -11.2 ug/m3. Which one is correct?

Response: Thank you for raising up this question. This study (MICS-Asia III) used three observational datasets: EANET, API, and Ref to evaluate the $PM_{10}$, and HTAP II used only EANET and API datasets. To make a fair comparison, we recalculated the MICS-Asia III multi-model performance at EANET and API sites. The MB in $PM_{10}$ is changed from -11.2 ug/$m^3$ to -18.6 ug/$m^3$.

**2Q20: Comment:** Line 240: Change "so-call" to "so-called".

Response: We have made the change in the manuscript.

**2Q21: Comment:** Lines 258-259: It is difficult to understand why it is so hard to get access to the emissions used by different groups. Using model simulated CO and BC as surrogates for emissions is not a good approach because they are strongly influenced not only by emissions but also by transport, deposition, and chemistry (in case of CO). Use of column CO is especially concerning because CO distribution in the free troposphere is primarily controlled by transport (inflow from domain boundaries in case of regional models) and not by emissions. Similarly, regions downwind of strong emission sources will exhibit disproportionately larger concentrations of BC and CO than the emissions. Therefore, I recommend using actual emission fluxes in your analysis in Section 3.2.

Response: We agree with the reviewer than CO and BC are not appropriate to indicate distributions of emission. Model treatments have caused differences in the model inputs (see the answer for Q1). Thus, we decided to compare indicators instead of direct model outputs in this manuscript. For this reason, section 3.2 is removed from the manuscript.

**2Q22: Comment:** Lines 325-326: If the models were able to capture $SO_2$ concentrations, would they have overestimated $SO_4$ considering that they are simulating $SO_4$ reasonably well even with underestimated $SO_2$ concentrations.

Response: As shown in the following figure, the SOR values of models are generally higher than the observation at sites E42, E43 and E44. This high SOR values reveal that models have converted more $SO_2$ to $SO_4^{2-}$ than the actual condition. In this case, it is possible that $SO_4^{2-}$ concentrations are overestimated while $SO_2$ concentrations are well fitted to observations.

**2Q23: Comment:** Line 364: Sea-salt emissions are controlled via a namelist option in WRF-Chem. Was there a specific reason for turning off sea-salt emissions in M7 and M8?

Response: We have confirmed with the model groups that sea-salt emissions are not included in the simulations by M7 and M8.

**2Q24: Comment:** Line 390: µg cm-3 or µg m-3?

Response: We have changed to µg m$^{-3}$ and kept it consistent in the manuscript.

[revised manuscript text omitted]

**Supplementary Material**

**Figures**

**Figure S1**

[Figure]

[Figure]

**Gas-aerosol parititioning of S in southern EA**
**Gas-aerosol parititioning of N**
Figure S1 Gas-particle conversion of S and N from observation and individual models at EANET sites. The unit is
µmole (S or N) m$^{-3}$. Values are calculated by annual average data.

**Figure S2**

[Figure]

Figure S2 The inter-model variations of PM and components among models. The 1sd is the 1 standard deviation
among models ($\mu g\ m^{-3}$). The 1sd% is calculated by dividing 1sd by MMM (%).

---

## Author Comment (AC2) · 15 Feb 2020

**Received and published: 21 November 2019**

General Comments: The manuscript examines the spatial and temporal variability in aerosol concentration and composition simulated by 12 regional models. The model predictions are evaluated with measurements from the different monitoring networks. It also quantifies the ensemble mean bias through several processes, namely model inputs, gas to particle conversion, the impact of sea-salt and dust emissions parameterization, as well as deposition processes. The topic and overall approach fit with ACP. However, I feel that there are some aspects need to be discussed in more detail and the paper needs some improvement before being published in ACP.

As Reviewer #2 already mentioned, the supplementary material contains a very large number of figures that could easily be added in the main text. In the current version, the readers have to go back and forth from the main text to the supplementary material.

It would be nice to have a more in-depth investigation of possible causes that could explain the substantial differences between simulations. As an example, there are large differences between simulations performed using the same model (see Figs.2, S9, S12 and S13). The M4 and M5 models seem to share similar setup and use the same input data, except the organic chemistry treatment. Then, how could the authors explain the different modelled spatial distribution of black carbon, SOR or  $C(NO_2)$ , as well as the different modelled concentrations of  $NO_2$  and  $NO_3$ ?

The logical connection between the manuscript and the SM needs to be improved. For example, the discussion associated with Fig. S8 appears after the one associated with Fig. S12.

Response: We would like to thank the reviewer for the valuable comments. Following are the responses to the comments.

**Specific comments**

3Q1: Lines 104-105: it repeats information already provided in the introduction Response: We have revised the manuscript and deleted the redundant information. 3Q2: Line 115: use "IC/BC" instead "BC"

Response: We have changed it.

3Q3: Lines 164-165: Why the online calculation of natural emissions of dust and sea salt particles fully embedded within the WRF-Chem is not included?

Response: We have confirmed with the modelling groups that the two WRF-Chem models did not include dust and sea-salt emissions in the simulations.

3Q4: Line 185: move this information into the main text

Response: We updated the model performance on aerosols in our companion paper (Chen et al., 2019). We removed the section of model evaluation from this manuscript and focused more on the inter-model comparison.

We also moved the figures and tables from the supplementary material to the main text to make it easier for the readers.

Reference: Chen, L., Gao, Y., Zhang, M., Fu, J. S., Zhu, J., Liao, H., Li, J., Huang, K., Ge, B., Wang, X., Lam, Y. F., Lin, C.-Y., Itahashi, S., Nagashima, T., Kajino, M., Yamaji, K., Wang, Z., and Kurokawa, J.: MICS-Asia III: multi-model comparison and evaluation of aerosol over East Asia, Atmos. Chem. Phys., 19, 11911–11937, https://doi.org/10.5194/acp-19-11911-2019, 2019.

3Q5: Lines 185-186: briefly describe the main findings of Chen et al., 2019

Response: We added the description on the main findings of Chen et al., 2019 as follows:

Lines 176-188: "Evaluation of model performance on aerosols can be found in our companion paper (Chen et al., 2019). The following are the main findings:  $PM_{10}$  concentrations were generally underestimated over the simulation domain.  $PM_{2.5}$  concentrations were also underestimated over Eastern EA but were well simulated in Central EA. Models failed to reproduce the high peaks of  $SO_4^{2-}$  concentration in Central EA, probably due to missing  $SO_4^{2-}$  formation mechanisms (such as

heterogeneous  $SO_4^{2-}$  chemistry), which has been reported as an important formation pathway of  $SO_4^{2-}$  in China.  $NO_3^{-}$  concentrations were overpredicted by most models over the simulation domain and were associated with the underestimation of  $SO_4^{2-}$ . M7 and M8 models produced significantly lower  $NO_3^{-}$  concentrations than observations and other models, due to model bias in simulating the  $NH_3$  concentrations and missing the  $N_2O_5$  heterogeneous reaction that severs as an import formation pathway of  $NO_3^{-}$ . The spatial distributions of AOD was generally well simulated except underestimation around the Himalaya mountains, Taklamakan Desert and Gobi Desert."

3Q6: Lines 191-193: should be an NMB of -300%?

Response: We removed section 3.2 from the manuscript (reasons are listed in our response to Q9). This sentence has been removed from the manuscript.

**3Q7: Lines 196-197: Fig. S3 instead of Fig. S2?**

Response: Here we want to refer to the location of HBT in figure S2 (now figure 1 in the revised manuscript).

3Q8: Lines 205-208: For northern and south-eastern EA these findings are not applicable. Could the authors explain why this behaviour is seen only for eastern EA, but not for the other analysed regions?

Response: On lines 205-208 (manuscript before revision), we described the model performance on simulating the trends of  $SO_4^{2-}$  and  $NO_3^{-}$  concentrations in eastern EA. Model underestimations in several months caused the correlations between model ensemble and observation to be low. This is also the reason for low R values in simulating  $SO_4^{2-}$  and  $NO_3^{-}$  in northern EA and southern EA.

This sentence has been removed from the manuscript.

3Q9: Lines 258-260: As Reviewer #1 already mentioned, using CO and BC concentration as proxies for emissions won't give a good indication of the emission's spatial distribution.

Response: After checking with the modelling groups, we notice that all models used the same anthropogenic and natural emissions. However, mismatch during the temporal and vertical treatment of emission files by different modelling groups have caused differences in the model inputs (Itahashi et al., 2019). In addition, different model set-up such as the heights of first vertical layer and spatial resolutions of model grids also affect the inter-model comparison on direct model outputs such as surface PM concentrations. Therefore, we decided to use indicators (such as SOR) instead of direct model outputs to facilitate comparison on model mechanisms. For this reason, we removed our discussion on the impacts of emissions and IC/BC on model performance (section 3.2 in the old manuscript) and focused on the comparison of the three processes: gas-aerosol portioning, dust mechanism and wet and dry deposition efficiency.

Reference: Itahashi, S., Ge, B., Sato, K., Fu, J. S., Wang, X., Yamaji, K., Nagashima, T., Li, J., Kajino, M., Liao, H., Zhang, M., Wang, Z., Li, M., Kurokawa, J., Carmichael, G. R., and Wang, Z.: MICS-Asia III: Overview of model inter-comparison and evaluation of acid deposition over Asia, Atmos. Chem. Phys. Discuss., https://doi.org/10.5194/acp-2019-624, in review, 2019.

3Q10: Lines 266-269: the models indeed do not provide the layer height in meters, but this can be easily calculated

Response: Thank you for the suggestion. As the reviewer mentioned in Q9, CO is not an appropriate indicator to reveal the vertical distributions of emissions, we removed the discussion and figure related to CO vertical distribution from the manuscript. However, discussion on the inter-model differences on the vertical layers can be found in our companion study (Li et al., 2019).

Reference: Li, J., Nagashima, T., Kong, L., Ge, B., Yamaji, K., Fu, J. S., Wang, X., Fan, Q., Itahashi, S., Lee, H.-J., Kim, C.-H., Lin, C.-Y., Zhang, M., Tao, Z., Kajino, M., Liao, H., Li, M., Woo, J.-H., Kurokawa, J., Wang, Z., Wu, Q., Akimoto, H., Carmichael, G. R., and Wang, Z.: Model evaluation and intercomparison of surface-level ozone and relevant species in East Asia in the context of MICS-Asia Phase III – Part 1: Overview, Atmos. Chem. Phys., 19, 12993– 13015, https://doi.org/10.5194/acp-19-12993-2019, 2019.

3Q11: Lines 273-276: these differences represent averaged values over the domain? It is difficult to connect these numbers with Fig S11. What do you mean by 2-6 ug m-3 for NO3? Do these numbers represent the range of differences? From Fig, S11 the range seems to be between -4 and 6 ug m-3.

Response: The numbers represent the ranges of values over the continental regions. For  $NO_3^-$ , the range is -4 to 6  $\mu$ g m-3.

This sentence has been removed from the manuscript.

3Q12: Lines 277-279: have you compared these values with observations?

Response: Lines 277-279 explain the differences in PM components around the edge of the simulation domain between the M1 and M2 is caused by using different boundary condition as model input. M1 used downscale results from GEOS-Chem and thus considered the impacts of emission and pollutions from outside of the research domain. M2 used the default values in the CMAQ model as a boundary condition, which is much smaller than those from GEOS-Chem. This result is not comparable to observations.

This sentence has been removed from the manuscript.

3Q13: Lines 297-300: Why? A different aerosol treatment?

Response: Yes, there is an update in the formation pathway of  $SO_4^{2-}$  particles in CMAQv502. We added the following sentences in the manuscript for explanation:

Lines 229-234: "The X-CMAQ models (including WRF-CMAQ and RAMS-CMAQ) produce similar *SOR* patterns, except that the CMAQv5.0.2 models (M1 and M2) predict 10% higher

*SOR* in the HBT region than the CMAQv4.7.1 models (M4, M5, and M6). CMAQv502 updated the production of  $SO_4^{2-}$  in the aqueous reaction of the older version (Appel et al., 2013). The explicit treatment of Fe and Mn allows more consistent treatment of aqueous reaction from  $SO_2$  to  $SO_4^{2-}$ ."

3Q14: Figs. 2 and S13 and subsequent discussion: Could the authors provide an explanation for the really low NO3 concentration modelled with M8? Have they considered the M8 model when the ensemble mean was compared against observation? Using such outliers, the MMM mean can even be deteriorated compared to individual simulations.

Response: The reasons have been discussed on page 11920 of our companion paper (Chen et al., 2019). One possible reason is the low production of  $NH_3$  by M8 (Fig S4 of Chen et al., 2019). Another reason is not including the  $N_2O_5$  heterogeneous reaction in M8, which is an important pathway for the formation of  $NO_3^-$  (Chen et al., 2019).

Yes, M8 is included in the ensemble mean. We agree if the MMM is used to show the model performance, the  $NO_3^-$  of M8 should be taken out. However, this paper aims at identifying the differences among models. The MMM is used as a reference to evaluate the differences between models. Thus, we kept including M8 in the MMM.

3Q15: Fig. 2 c) M3 should be M4

Response: We have modified the figure.

3Q16: Lines 323-330: if the S emissions were insufficient, why the M13 and M14 models provide reasonable results?

Response: Our current analysis can only demonstrate that the SOR values of models are generally higher than observations. Since model inputs are not identical in different models due to the reason we mentioned in our answer to Q9, we can't confirm if the underestimation of SO2

is caused by insufficient S emissions. For this reason, we removed the related discussion from the manuscript.

3Q17: Section 3.4: same as before, why not using the online calculation of natural emissions of dust particles within WRF-Chem?

Response: We have confirmed that the two WRF-Chem models do not include dust emissions during the simulation.

3Q18: Line 374: "perfectly" is a strong word

Response: We have changed the word to "reasonably".

3Q19: Lines 432-433: Table 4 and the associated discussion. Please be consistent when calculating the total wet and dry deposition of S/N and doing subsequent analysis.

Response: We have modified the manuscript and checked the calculation to be consistent and clear.

3Q20: Lines 460-473: In the case of wet deposition of S and N, M11 shows a very different spatial pattern compared with the other models. What is the reason for this? It would be nice if the authors will try to analyse in detail the causes of these differences.

Response: We added the explanation in the manuscript as follows:

Lines 403-405: "Both  $\lambda_{wet}$  and  $V_d$  of M11 are much lower than the other models, especially over eastern EA. And this is a possible reason for the biased performance of M11 on wet deposition (Fig. 7)"

1 Why models perform differently on particulate matter over East

**2 Asia? – A multi-model intercomparison study for MICS-Asia III**

Jiani Tan1, Joshua S. Fu1, Gregory R. Carmichael2, Syuichi Itahashi3, Zhining Tao4, Kan

4 Huang1,5, Xinyi Dong1, Kazuyo Yamaji6, Tatsuya Nagashima7, Xuemei Wang8, Yiming Liu8,

5 Hyo-Jung Lee9, Chuan-Yao Lin10, Baozhu Ge11, Mizuo Kajino12, Jia Zhu11, Meigen Zhang11,

- 6 Hong Liao13 and Zifa Wang11

[revised manuscript text omitted]

---

## Editor Decision (ED1)

**Editor comments:** You have addressed the referee comments satisfactorily in your response. However, some of these responses should be also added to the manuscript. In addition, I have some minor/technical comments that should be considered before final acceptance.

Main comments

1) Referee #3 had asked about reasons for the low NO3 concentration predicted by M8. In the response, you wrote that NH3 production is very low. However, in the manuscript, you only mention that the N2O5 hydrolysis is missing (l. 193). Please add here and/or at a different appropriate place also the lower NH3 'production'. Please also clarify what you mean by this. I assume that you mean NH3 emission ('production' would imply a chemical formation) that leads to ammonia, which in turn, then leads to enhanced NH4NO3 levels (?)

2) Finally in the last response to the referee report, you added an explanation why the M7 and M8 did not include sea-salt emission ('turned off by mistake'). Since both referees were puzzled why the models did not include these emissions, this information should be added (e.g. in line 305). It is completely acceptable to admit in a paper that mistakes were made in the model set-up.

3) In the last referee report, Reviewer #2 commented on the discussion in lines 397ff (in the most recent manuscript version). In the response to the referee, you gave some very brief explanation why you prefer comparing the wash-out ratio rather than comparing C(depo). Please expand on this explanation and add it to the manuscript.

4) In the last referee report, Reviewer #2 suggested that also uncertainties on OH and/or ozone concentration may lead to uncertainties in the predicted gas-aerosol conversion of S and N. Only in the conclusion section, you vaguely mention this possibility (l. 430 ff). Are there any previous model studies that discuss such uncertainties? In any case, possible uncertainties in the oxidant levels should already discussed earlier, in Section 3.2.

Minor/technical comments

Add units to all parameters used in the equations.

l.  29/30 and 41/42: This text is repetitive. Please reword or remove accordingly.

l. 132: 'Particles participate in the cloud condensation nuclei' should be reworded.

l. 171: replace 'satisfied' by 'satisfying'

l. 174: replace 'intensively located' by 'concentrated'

l. 193: replace 'sever as an import formation process' by 'serve as an important formation process'

l. 212: replace 'process' by 'processes'

l. 212 and 214: replace 'group' by 'groups'

l. 213: replace 'but' by 'a'

l. 283: '…the differences between the two are smaller' – please clarify: smaller than what?

l. 301: replace 'comes' by 'come'

l. 366: replace 'succeed' by 'succeeds'

---

## Author Response (AR2)

Review of "Why models perform differently on particulate matter over East Asia? – A multi-model intercomparison study for MICS-Asia III" by Tan et al.

I thank the authors for revising their manuscript in response to my comments. However, the authors did not address all the comments adequately as explained below.

Response: We appreciate the reviewer's kindness in reviewing the manuscript and providing the valuable comments. Following are the point-to-point responses to the comments.

For example, in response to my comment on "Why would natural emissions affect BC concentrations and CO loading?", the authors state that same anthropogenic and natural emissions were used for each modeling system. But natural emissions cannot be same as they are calculated online in most of the models and some of the participating models do not even turn on the natural emissions.

Response: Thank you for pointing out this question.
The sea-salt and dust emissions were not prescribed. Modelling groups used their own sea-salt and dust emissions produced by the modules in the models. The emission group of MICS-Asia III prepared the biogenic emissions with Model of Emissions of Gases and Aerosols from Nature (MEGAN), biomass burning emissions with Global Fire Emissions Database (GFED), volcanic $SO_2$ emission from AEROCOM program (https://aerocom.met.no/ DATA/download/emissions/AEROCOM_HC/volc).

We made the following changes in the manuscript:
Line 110-112: "All modelling groups are required to use the prescribed anthropogenic emissions and natural inputs (including biogenic emissions, biomass burning emissions and volcanic $SO_2$ emissions. Dust and sea-salt emissions are produced by the corresponding modules in the models)."

I also asked to clarify "why sea-salt emissions were turned off in M7 and M8?". In their response, the authors say they confirmed that sea-salt emissions were turned off (which was clear in the previous version) but did not give a reason to turn them off. Please clarify.

Response: Model results of M7 and M8 were submitted by two different participating modelling groups of MICS-Asia III. They turned off the sea-salt emissions by mistake.

Uncertainty in Gas-particle conversion is suggested as the main reason for inter-model differences. Are the differences due to differences in tropospheric ozone and thus simulated OH or are the parameterization converting sulfuric acid to sulfate are different?

Response: Yes, both two factors could affect the gas-particle conversion. We added the following discussion in the manuscript:

Lines 428-431: "Besides the inter-model differences in the pathways of $SO_4^{2-}$ and $NO_3^-$ formation, the abundance of oxidants (i.e. OH radical) also affects the gas-aerosol conversion of S and N. In addition, the conversion between sulfuric acid and $SO_4^{2-}$ depends on the abundance of neutralizers such as $Na^+$ and $NH_4^+$."

In addition, I have the following minor comments.

Line 31: Change "one the" to "one of the"

Response: We have changed it in the manuscript.

Line 122: Change Sector 2.1 to "Section 2.1".

Response: We have changed it in the manuscript.

Line 154: Should not Japan and Korea classify eastern EA?

Response: It is a typo. Japan and Korea are classified in eastern EA in the analysis. We have changed it and checked the manuscript to be consistent.

Lines 176-188: Can you please provide some quantitative information like by how much $SO_4$ is overestimated and $NO_3$ is underestimated. Same information for other species and AOD.

Response: We added the following sentences in the manuscript:

Line 180-183: "the differences between MMM and observation/satellite data for the surface concentrations of $PM_{10}$, $PM_{2.5}$, $SO_4^{2-}$, $NO_3^-$ and $NH_4^+$, and column integrated aerosol optical depth (AOD) were -32.6%, 4.4%, -19.1%, 4.9%, 14.0% and 18.7%, respectively (calculated with normalized mean biases (NMBs))."

Lines 389-404: I am puzzled by this discussion. It is not clear to me how lower values of washout ratio lead to large biases in M11. I think showing the spatial distribution of Cdepo will be more useful than showing the washout ratios.

Response: The amount of deposition ($C_{depo}$) is determined by the surface concentration of air pollutants ($C_{surface\_air}$) and the washout ratio (also called scavenging efficiency,

determined by the model mechanism in producing wet deposition). Thus, inter-model differences in direct model outputs of $C_{depo}$ may be partial influenced by different model inputs, caused by mismatch occurred in vertical and temporal allocation of emission inputs and employment of different mechanisms to produce dust and sea-salt emissions. To avoid such impacts, we used the washout ratio, calculated as a ratio of $C_{depo}$ to $C_{surface\_air}$, as an indicator to reveal the inter-model differences caused by model mechanisms.

---

## Author Response (AR3)

**Editor comments:** You have addressed the referee comments satisfactorily in your response. However, some of these responses should be also added to the manuscript. In addition, I have some minor/technical comments that should be considered before final acceptance.

Main comments
1) Referee #3 had asked about reasons for the low $NO_3$ concentration predicted by M8. In the response, you wrote that $NH_3$ production is very low. However, in the manuscript, you only mention that the $N_2O_5$ hydrolysis is missing (l. 193). Please add here and/or at a different appropriate place also the lower $NH_3$ 'production'. Please also clarify what you mean by this. I assume that you mean $NH_3$ emission ('production' would imply a chemical formation) that leads to ammonia, which in turn, then leads to enhanced $NH_4NO_3$ levels (?)
Response: Both $NH_3$ and $NH_4^+$ of M8 were lower than the other models according to Fig. 3 and Fig. S4 of Chen et al., 2019. It is very likely that M8 used low $NH_3$ emission, which led to less conversion of $NH_3$ to $NH_4^+$ and prevented the formation of $NH_4NO_3$.

We have modified the following sentences in lines 189-192:
"M7 and M8 models produced significantly lower $NO_3^-$ concentrations than observations and other models, due to underestimation in $NH_3$ concentrations (might be caused by low $NH_3$ emission) and missing the $N_2O_5$ heterogeneous reaction that sever as an important formation pathway of $NO_3^-$ (Chen et al., 2019)."

2) Finally in the last response to the referee report, you added an explanation why the M7 and M8 did not include sea-salt emission ('turned off by mistake'). Since both referees were puzzled why the models did not include these emissions, this information should be added (e.g. in line 305). It is completely acceptable to admit in a paper that mistakes were made in the model set-up.
Response: We modified the description about the sea-salt emission of M7 and M8 in line 309 as follows:
"In this study, the WRF-Chem models (M7 and M8) turned off the sea-salt emissions, thus their PMC concentrations over the oceans and seas are not defined."

3) In the last referee report, Reviewer #2 commented on the discussion in lines 397ff (in the most recent manuscript version). In the response to the referee, you gave some very brief explanation why you prefer comparing the wash-out ratio rather than comparing C(depo). Please expand on this explanation and add it to the manuscript.
Response: The two indicators: $\lambda_{wet}$ and $V_d$ were first mentioned in lines 133-148, where we introduced the mechanism of wet and dry deposition.
We added the following sentences in lines 401-408 to explain the reasons for using these two indicators instead of direct model outputs of wet and dry deposition.
"The amount of wet deposition is determined by the $C_{surface\_air}$ and $\lambda_{wet}$ (mentioned in sec. 2.2). And in this study, $C_{surface\_air}$ may be partial influenced by different model inputs, caused by mismatch occurred in vertical and temporal allocation of emission inputs and employment of different mechanisms to produce dust and sea-salt emissions. Thus, we

used $\lambda_{wet}$, instead of direct model outputs of wet deposition, as an indicator to reveal the inter-model differences on wet deposition in the following analysis. For the same reason, we used $V_d$ as an indicator for inter-model comparison on dry deposition."

4) In the last referee report, Reviewer #2 suggested that also uncertainties on OH and/or ozone concentration may lead to uncertainties in the predicted gas-aerosol conversion of S and N. Only in the conclusion section, you vaguely mention this possibility (l. 430 ff). Are there any previous model studies that discuss such uncertainties? In any case, possible uncertainties in the oxidant levels should already discussed earlier, in Section 3.2.
Response: We moved the discussion on the impacts of OH on gas-aerosol conversion of S and N from the conclusion section to sect. 3.2 (lines 288-293). We expanded the discussion and cited previous studies as follows:
Lines 288-293: "Besides the inter-model differences in the pathways of $SO_4^{2-}$ and $NO_3^-$ formation, the interaction between aerosols and atmospheric oxidants can also affect the formation of aerosols (Liao et al., 2003). Aerosols affect the tropospheric oxidants (i.e. $HO_x$) budget by altering the photolysis rates and uptake of reactive gases (Tie et al., 2003; Li et al., 2018). In turn, the abundance of $HO_x$ affects the gas-aerosol conversion of S and N. In addition, the conversion between sulfuric acid and $SO_4^{2-}$ depends on the abundance of neutralizers such as $Na^+$ and $NH_4^+$."

Minor/technical comments
Add units to all parameters used in the equations.
Response: We have added units to all parameters in the equations.

l. 29/30 and 41/42: This text is repetitive. Please reword or remove accordingly.
Response: We deleted the repetitive sentences in lines 41-42.

l. 132: 'Particles participate in the cloud condensation nuclei' should be reworded.
Response: We have reworded the sentence as follows:
"Particles take part in the cloud condensation nuclei"

l. 171: replace 'satisfied' by 'satisfying'
Response: We have replaced the words.

l. 174: replace 'intensively located' by 'concentrated'
Response: We have replaced the words.

l. 193: replace 'sever as an import formation process' by 'serve as an important formation process'
Response: We have replaced the words.

l. 212: replace 'process' by 'processes'
Response: We have replaced the words.

l. 212 and 214: replace 'group' by 'groups'

Response: We have replaced the words.

l. 213: replace 'but' by 'a'
Response: We have replaced the words.

l. 283: '…the differences between the two are smaller' – please clarify: smaller than what?
Response: We changed the sentence to "…the differences between the two in $C(NO_2)$ are smaller than those in *SOR*."

l. 301: replace 'comes' by 'come'
Response: We have replaced the words.

l. 366: replace 'succeed' by 'succeeds'
Response: We have replaced the words.